# DEATH OF THE NOVEL(TY): BEYOND $n$-GRAM NOVELTY AS A METRIC FOR TEXTUAL CREATIVITY

**Arkadiy Saakyan**
Columbia University

**Najoung Kim**
Boston University

**Smaranda Muresan**
Columbia University

**Tuhin Chakrabarty**
Stony Brook University

`a.saakyan@cs.columbia.edu`

## ABSTRACT

$N$-gram novelty is widely used to evaluate language models' ability to generate text outside of their training data. More recently, it has also been adopted as a metric for measuring textual creativity. However, theoretical work on creativity suggests that this approach may be inadequate, as it does not account for creativity's dual nature: novelty (how original the text is) and appropriateness (how sensical and pragmatic it is). We investigate the relationship between this notion of creativity and $n$-gram novelty through 8,618 expert writer annotations of novelty, pragmaticality, and sensicality via *close reading* of human- and AI-generated text. We find that while $n$-gram novelty is positively associated with expert writer-judged creativity, approximately $91\%$ of top-quartile $n$-gram novel expressions are not judged as creative, cautioning against relying on $n$-gram novelty alone. Furthermore, unlike in human-written text, higher $n$-gram novelty in open-source LLMs correlates with lower pragmaticality. In an exploratory study with frontier closed-source models, we additionally confirm that they are less likely to produce creative expressions than humans. Using our dataset, we test whether zero-shot, few-shot, and finetuned models are able to identify expressions perceived as novel by experts (a positive aspect of writing) or non-pragmatic (a negative aspect). Overall, frontier LLMs exhibit performance much higher than random but leave room for improvement, especially struggling to identify non-pragmatic expressions. We further find that LLM-as-a-Judge novelty ratings align with expert writer preferences more so than an n-gram based metric. [1]

## 1 INTRODUCTION

Advances in large language models (LLMs) have led to their widespread applications in writing. In fact, recent studies (Handa et al., 2025; Chatterji et al., 2025) show that writing assistance remains one of the main use cases of LLMs. At the same time, researchers have been raising concerns about how writing assistance tools can reduce collective human creativity via homogenization effects (Doshi & Hauser, 2024; Kobak et al., 2025; Zhang et al., 2025), proliferation of AI slop (Chakrabarty et al., 2025a; Shaib et al., 2025) or copying from training data (McCoy et al., 2023).

Such challenges lead to a growing need for robust textual creativity evaluation. Recently, tools like WiMBD (Elazar et al., 2024), Rusty-DAWG (Merrill et al., 2024), infini-gram(-mini) (Liu et al., 2024; Xu et al., 2025) have been developed to efficiently search LLMs' pretraining corpora and evaluate the novelty of their generations. Building on these tools, Lu et al. (2025) introduced a metric called CREATIVITY INDEX, which places significant weight on $n$-gram novelty – lack of occurrence of textual fragments in some large (several trillion tokens) corpora – for measuring creativity of text. However, literature on the psychology of creativity would consider such an approach as not fully adequate: based on the widely adopted definitions of creativity, novelty is a necessary but not sufficient criterion (Sawyer & Henriksen, 2024; Runco & Jaeger, 2012; Csikszentmihalyi et al., 1997; Amabile, 1983; Jackson & Messick, 1965). An expression generated by an LLM may be novel with respect to the pretraining corpora but make little sense (e.g., *stitched the frayed edges of cloaks and ragged seams of fear* in Figure 1), precluding it from being judged as creative. At the same time, an expression may not be $n$-gram novel with respect to pretraining but still be creative. For instance, *that's the bottom of the heart, where blood gathers* has low $n$-gram novelty, but was

---

[1] We will release the collected dataset and models on `github.com/asaakyan/ngram-creativity`.

rated as creative in the context of the given passage by an expert writer because of how it comes across as emotionally foreshadowing (see Table 1).

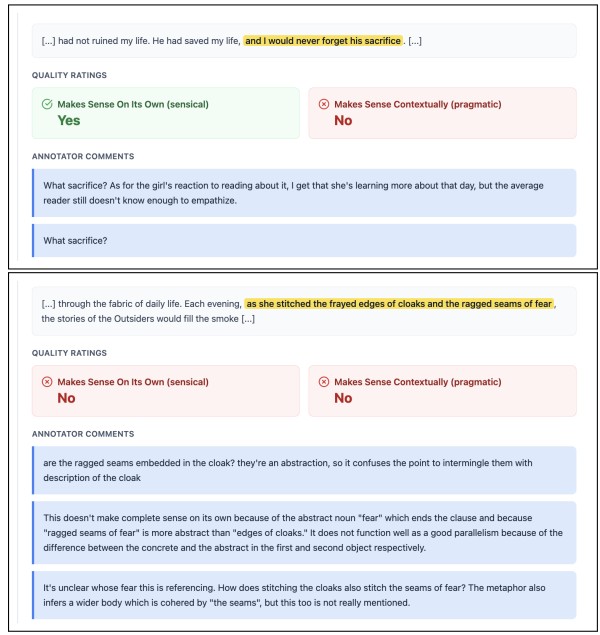

Figure 1: Examples of non-pragmatic expressions (top), and expressions that are both non-sensical and non-pragmatic (bottom).

While prior work (Chakrabarty et al., 2024b) has relied on expert human evaluation for measuring creativity of long form text via fine-grained rubrics, LLMs struggle to imitate nuanced expert judgments, making this approach difficult to scale and automate. Instead, we turn to the standard definition of creativity (Runco & Jaeger, 2012), operationalizing it as a combination of novelty and appropriateness. We further decompose appropriateness into sensicality (making sense on its own) and pragmaticality (making sense in context). Figure 1 shows an example of a sensical but non-pragmatic expression (suddenly mentioning *sacrifice* at the end of the passage without providing appropriate context).

To formalize the failure modes of novelty as a metric, we conduct a study on the relationship between $n$-gram novelty – low occurrence in trillion token-level corpora – and creativity. Expert writers were asked to *close read* AI- and human-written passages and provide expression-level sensicality, pragmaticality, and perceived novelty ratings. Mixed-effects regression model analysis accounting for individual rater variation and other confounding factors revealed a negative relationship between $n$-gram novelty and pragmaticality in open-source models, with $\approx 91\%$ of highly $n$-gram novel expressions not judged as creative. This cautions against relying on $n$-gram novelty alone to evaluate creativity. To understand whether LLM-as-a-Judge frameworks could be used instead, we use the collected dataset to evaluate how well frontier LLMs and finetuned models can emulate expert human judgments. We further validate our best performing model on out-of-distribution data, finding a strong alignment between the LLM-as-a-Judge novelty scores and expert preferences.

## 2 DATA

**OPERATIONALIZING CREATIVITY** We operationalize creativity through appropriateness and perceived novelty components (Runco & Jaeger, 2012). We decompose appropriateness into two subcomponents: *sensicality* and *pragmaticality*. Sensicality is whether the expression makes sense standalone (for example, an expression like "*he tended at cloud finger*" does not make sense by itself, or in other words, semantically infelicitous). Pragmaticality refers to the expression's fit within the context of the passage, capturing a range of errors, including logical incoherence (e.g., "*Alice felt great*" immediately followed by "*Despite being sad, Alice...*"), or sounding awkward or odd in that context.[2][3] In addition, we collect *perceived novelty* ratings by asking whether expression is unusual, surprising or otherwise original[4]. *Creative expressions are those that are simultaneously judged by a human as sensical, pragmatic, and novel.* Text that is creative in this sense may not necessarily be $n$-gram novel (see Table 1).

---

[2]Note this allows for deliberate violations of pragmatic norms, e.g. in surreal writing.

[3]Pragmaticality is a more stringent requirement than sensicality (an expression cannot be pragmatic if non-sensical). We make this distinction to analyze more nuanced effects of context rather than atomic expression-level non-sensicality.

[4]Our annotations suggest that pragmaticality is a precondition for human-judged novelty, as discussed later.

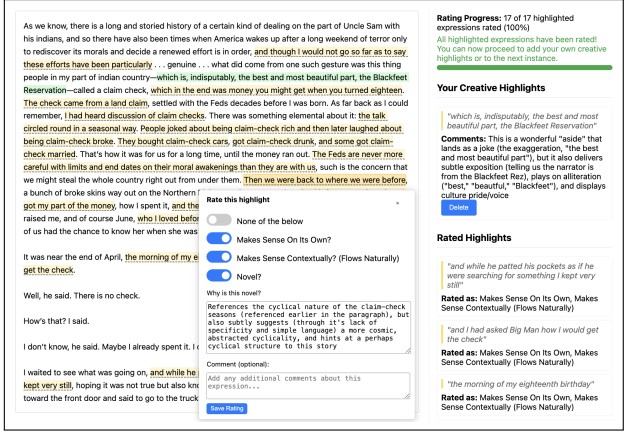

Figure 2: Example of the annotation interface and an expert writer's annotation.

**SURVEY INSTRUMENT** To obtain the annotations, we took inspiration from the practice of *close reading* (Smith, 2016), a literary analysis technique of carefully examining small parts of the passage to make sense of it as a whole, "putting the author's choices under a microscope" (Brummett, 2018). We first split the passage by punctuation into "atomic" expression units, and then sample roughly 50% of them for annotation based on the percent of novel $n$-grams within the expression (see Appendix A.1). These expressions are *pre-highlighted*. Then, we ask the annotators to rate each of the pre-highlighted expressions in the passage for their sensicality, pragmaticality, and novelty (see the user interface in Figure 2). In addition, the interface allows experts to highlight any creative expressions (which satisfy the sensicality, pragmaticality and novelty requirements) that were not already pre-highlighted. The annotators were required to provide rationales for why the expression was creative or perceived novel, and could leave an optional comment on other aspects. We provide detailed instruction and examples to the annotators with simplified terms for sensicality ("Makes sense on its own?") and pragmaticality ("Makes sense contextually? (Flows naturally)")—see Appendix A.3. An example of an annotation is shown in Figure 2, where the pop-up window displays the annotation of a pre-highlighted expression, and in green the annotator highlighted a creative expression.

**RECRUITMENT** We recruit professional writers ($n = 26$) via listservs of top Master of Fine Arts writing programs in the US, as well as published writers with a Masters in English Writing or Literature through the Upwork platform. Each annotator first underwent training on at least one passage during which any instruction misunderstandings were corrected. Annotators were paid $\approx$ 100 USD per annotated batch of 10 passages, which took approximately 2–3 hours.

**ANNOTATION** The fiction domain is best suited for evaluating textual creativity, as it does not impose factuality constraints unlike technical or journalistic writing, and is less subjective in creativity evaluation than poetry (Kaufman et al., 2008; 2009). Hence, for seed human data we collect 50 passages of $\approx$ 400 words each from fiction published in the New Yorker.[5] We use fully open-source (both weights and training data) LLMs: OLMo (7B) (Groeneveld et al., 2024) and OLMo-2 (32B) (Team OLMo et al., 2025) to generate LLM passages, considering both the older and the improved, larger models to understand whether progress in language modeling and benchmark performance led to improvements in writing quality. The passages were chosen such that they are not present in the corresponding open-source LLM training corpora (see Appendix A.2). We assign each model 25 passages from the human sample (*seed passages*), and prompt them to generate a similar length passage based on the content obtained from the summary of the human text (see details in Appendix C.2). To ensure reasonable completion time for each annotator task, we divide the total 100 passages into 10 batches of 10 passages each, containing 5 human and 5 AI passages. Three annotators are assigned per batch. To gauge the ceiling LLM performance, we conduct a smaller-scale follow-up study[6] on 2 frontier models: GPT-5 and Claude 4.1, each generating 5 passages to make up one additional batch of 10 texts assigned to 4 randomly selected annotators from our pool.

**STATISTICS** Overall for the main and frontier model study, 8618 pre-highlighted expression annotations were collected for 2783 unique expressions, with 589 unique expressions rated as perceived novel by at least one annotator, 722 as non-pragmatic and 274 as non-sensical. Pragmaticality and sensicality were largely preconditions for perceived novelty judgments, as only 3% and 5% of perceived novel annotations were also marked as non-sensical and non-pragmatic, respectively. In addition to the pre-highlighted expression ratings, the annotators highlighted 241 new expressions as

---

[5]One of the most prestigious literary magazines for publishing fiction.

[6]Our main study focuses on LLMs for which pretraining corpora is known and hence the only models that allow accurate $n$-gram novelty estimation.

creative (satisfying the sensicality, pragmaticality, and perceived novelty criteria at the same time). Due to low prevalence of expressions annotated as perceived novel (7%) and non-pragmatic (8%), traditional agreement metrics would suffer from prevalence bias and would not be suitable (Eugenio & Glass, 2004; Brennan & Silman, 1992; Feinstein & Cicchetti, 1990). Hence, we report the free-marginal multirater Kappa $\kappa_{free}$ (Randolph, 2005). Across the 10 batches, the mean $\kappa_{free}$ for perceived novelty was 0.78 (sd = 0.11), and for pragmaticality 0.72 (sd = 0.12).

## 3 METHODS

**MEASURING $n$-GRAM NOVELTY** We use the `infinigram` package (Liu et al., 2024) to estimate $n$-gram novelty with respect to LLM training corpora. Specifically, we use $\infty$-probability, which allows to assign a probability to any expression using backoff to the longest expression present in the corpus $\mathcal{D}$:

$$P_\infty(w_i \mid w_{1:i-1}) = \frac{\text{cnt}(w_{i-(n-1):i-1}w_i \mid \mathcal{D})}{\text{cnt}(w_{i-(n-1):i-1} \mid \mathcal{D})}$$

where $n = \max\{n' \in [1, i] \mid \text{cnt}(w_{i-(n'-1):i-1} \mid \mathcal{D}) > 0\}$. We can then use the consecutive probabilities to compute the perplexity of an expression as a proxy for $n$-gram novelty.[7] We use the respective OLMo and OLMo-2 training corpora[8] as the reference $\mathcal{D}$.

**MODELING** Annotation of textual creativity is inherently subjective. To account for annotator variation as well as other confounding factors (topic of the passage, generation model), we turn to hierarchical/multilevel modeling (Gelman & Hill, 2007) commonly used to handle nested data in behavioral and social sciences (Baayen et al., 2008; Yarkoni, 2022; Kaufmann et al., 2025) and account for confounders in language model evaluation (Lampinen et al., 2022). Since our target variables are binary judgments of sensicality, pragmaticality, and perceived novelty, we fit logistic regression models. For instance, to model the relationship between $n$-gram novelty (measured by perplexity) and creativity, we fit the following model for the probability of the expression being labeled as creative (i.e. sensical, pragmatic, and novel simultaneously):

$$\text{logit}\big(\mathbb{P}\left[\text{creative}_i = 1\right]\big) = \beta_0 + \beta_1 \cdot \texttt{ppl\_log\_std}_i + \beta_{\texttt{gs}[i]} \cdot \texttt{gs}_i + u_{r[i]}^{(\text{rater})} + u_{p[i]}^{(\text{para})}$$

where `ppl_log_std` is log-standardized (to reduce skew and ease interpretation) perplexity, `gs` is a categorical variable representing the generation source (human, OLMo, or OLMo-2), and $a[i]$, $p[i]$ are the indices of the rater and seed passage id for expression $i$. Variation in each rater's baseline threshold for rating something as creative is represented by the varying intercept $u_{r[i]}^{(\text{rater})}$, while variation in prevalence of creative expressions by different topics (seed passages) is represented by the varying intercept $u_{p[i]}^{(\text{para})}$. Random intercepts are assumed to be normally distributed with a mean of zero and their respective group variances. See detailed specifications in Appendix B.

## 4 RESULTS

**$n$-GRAM NOVELTY PREDICTS CREATIVITY BUT IS NOT A RELIABLE METRIC** We find that standardized log perplexity is significantly associated with creativity (OR (Odds Ratio) $\approx 1.96$ per SD (Standard Deviation), $p < 0.001$). However, $\approx 91\%$ ($n = 3928$, Wilson 95% CI: $[0.90, 0.92]$) of top-quartile $n$-gram novel expressions are *not* judged as creative (sensical, pragmatic, and novel at the same time), with $\approx 79\%$ ($n = 1384$, Wilson 95% CI: $[0.77, 0.81]$) of unique such expressions not judged as creative by *any* of the annotators. Further, a substantial portion of creative expressions have very low perplexity: $\approx 25\%$ ($n = 605$, Wilson 95% CI: $[0.21, 0.28]$) of unique creative expressions fall below the mean perplexity, and 8% ($n = 605$, Wilson 95% CI: $[0.06, 0.10]$) are in the lowest quartile. This demonstrates that while *high perplexity is predictive of creativity, a non-negligible fraction of creative expressions are not $n$-gram novel, and the vast majority of highly*

---

[7]Defined in a standard way as $\left(\prod_{i=1}^{N} P_\infty(w_i | w_{1:i-1})\right)^{-\frac{1}{N}}$

[8]`Dolma-v1.7` (2.6T tokens) for OLMo and `v4_olmo-2-0325-32b-instruct` (4.2T tokens) for OLMo-2.

*$n$-gram novel expressions are not creative, cautioning against relying solely on $n$-gram novelty to capture creativity.* In Table 1 we provide examples of expressions that were judged as creative by the annotators along with their justifications, but have a very low log-standardized perplexity (among the lowest in the dataset). Annotator justifications indicate the importance of contextual factors rather than $n$-gram novelty for judging an expression as creative (more examples in Appendix A.4).

**$n$-GRAM NOVELTY NEGATIVELY IMPACTS PRAGMATICALITY IN OPEN-SOURCE LLMS** We fit a logistic regression model on whether an expression was rated as pragmatic using standardized log perplexity, generation source, and their interaction as predictors, with varying intercepts for annotators and seed passage IDs. Instances that were labeled as not sensical were removed to focus on cases where expressions do not make sense *in the context*, rather than simply not felicitous standalone.

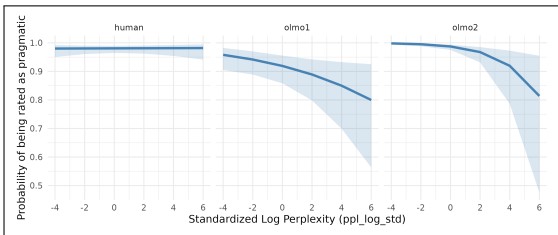

Figure 3: Predicted probability of being rated as pragmatic for different values of log-std perplexity, by generation source. Bands indicate 95% CIs. Annotator and paragraph intercepts correspond to population-level fixed effects.

Figure 3 demonstrates how $n$-gram novelty affects the probability of an expression being rated as pragmatic according to our model. We find negative slopes for both OLMo and OLMo-2 generated text, while we see no such effect for human-written text. Linear hypothesis tests show no evidence that $n$-gram novelty affects pragmaticality for human-written text ($\beta = 0.01$, SE $= 0.09$, $p = 0.92$). In contrast, higher $n$-gram novelty negatively affects pragmaticality of AI-generated text: OLMo ($\beta = -0.17, \chi^2(1) = 4.89, p = 0.027$) and OLMo-2 ($\beta = -0.48, \chi^2(1) = 13.80, p < 0.001$). This indicates that *as open-source LLMs try to generate more $n$-gram novel text, they are less likely to generate expressions that make sense contextually.* In Table 1, we show examples of highly $n$-gram novel expressions that were rated as non-pragmatic.

**AN EXPLORATORY STUDY WITH FRONTIER CLOSED MODELS** We conduct an exploratory follow-up study with two frontier models: GPT-5 and Claude-4.1 (see details in Appendices C.3, B.4).

| Contrast | Odds Ratio | 95% CI | $p$-value |
|---|---|---|---|
| Claude-4.1 / Human | 0.521 | [0.289, 0.939] | **0.024** |
| GPT-5 / Human | 0.511 | [0.279, 0.938] | **0.024** |
| OLMo / Human | 0.500 | [0.370, 0.676] | **<0.001** |
| OLMo-2 / Human | 0.588 | [0.439, 0.788] | **< 0.001** |

Table 2: Estimated marginal means (EMM) contrasts comparing creativity of AI and human expressions. $OR < 1$ indicate model-generated expressions were less likely to be judged creative compared to human. $p$-values adjusted for multiple comparisons using the method of Dunnett (1955).

Expanding on the original dataset, we compare the probability of expression creativity across generation sources (human, OLMo, OLMo-2, GPT-5, Claude-4.1) by fitting a logistic regression with generation source as the predictor and random intercepts for annotators and seed passages. Table 2 shows that *the probability of an expression to be judged creative is significantly higher for humans compared to frontier and open-source LLMs.* Although not statistically significant (likely due to small sample of the exploratory study and imprecise estimation of $n$-gram novelty as we have no access to the models' pretraining data), we observe a similar trend of a negative impact of $n$-gram perplexity when comparing frontier model and human expressions: the interaction terms between perplexity and generation source were negative for both Claude ($\beta = -0.22, p = 0.44$) and GPT-5 ($\beta = -0.17, p = 0.32$), but slightly positive for the human baseline ($\beta = 0.13, p = 0.49$). Future work could utilize our close reading annotation scheme to conduct a higher-powered study.

**LACK OF EVIDENCE THAT AI LIKELIHOOD PREDICTS NON-CREATIVITY** One plausible hypothesis is that if a text seems more AI-generated, it is judged by humans as less creative. We fit a logistic regression predicting the probability of the expression being judged creative or pragmatic

| *Expression* | Justification | PPL | Source |
|---|---|---|---|
| | — **Low $n$-gram Novelty but Creative** — | | |
| *Even when you're here, you're not here* | This is a creative way of showing the emotional disconnection of the narrator and how that is the source of problems with his partner. | -2.3 | human |
| *knowing that he wouldn't.* | I found this to be a creative expression because it offers insight into Tongsu's mind and surprises the reader with its blunt honesty. | -2.1 | human |
| *That's the bottom of the heart, where blood gathers.* | The metaphor is vivid and physiological, merging the act of writing with the body in a tactile, somber way. It's also emotionally foreshadowing. | -1.5 | human |
| | — **High $n$-gram Novelty but Non-Pragmatic** — | | |
| *and said the morning blessings in a whisper that embarrassed the chairs* | | 2.37 | GPT-5 |
| *the child who was at once a present fundamental fact but* | In isolation and in context, this is a misadventure of poor word choice, punctuation, and structure. | 2.36 | human |
| *each word spoken a thread in the slow weaving of trust* | [...] saying the act of weaving is made up of physical threads is mixing metaphors;[...] | 2.09 | OLMo-2 |
| *tape suspended between her fingers like a frozen gesture* | | 1.40 | Claude-4.1 |

Table 1: Top: expressions judged creative despite low $n$-gram novelty (negative log-standardized perplexity). Bottom: non-pragmatic expressions with high $n$-gram novelty (positive log-standardized perplexity).

given the AI likelihood scores from one of the leading AI-generated text detectors, Pangram (Emi & Spero, 2024). The detector perfectly classified AI texts assigning all of them a score of $100\%$; human texts were assigned varying small likelihoods of sounding like AI (see Figure 9). Log-standardized AI likelihood scores were used as a passage-level predictor in the logistic regression model with varying intercepts for annotators and passages. We did not find evidence that passages with higher (or lower) AI likelihood scores systematically differed in the average novelty or pragmaticality of their expressions. A one-standard deviation increase in a passage's AI likelihood was associated with a small, non-significant increase in the odds that an expression was judged creative ($\beta = 0.05$, SE $= 0.11$, $p = 0.624$) and a non-statistically significant decrease in the probability that it was judged pragmatic ($\beta = -0.25$, SE $= 0.15$, $p = 0.09$).

**WRITING QUALITY REWARD MODELS PREDICT CREATIVITY AND PRAGMATICALITY** Chakrabarty et al. (2025a) recently explored training scalar reward models on a corpus of <AI-generated, Expert-edited> pairs of text with implicit preference judgments (edited > original) (Chakrabarty et al., 2025b). Unlike for AI-likelihood scores, a logistic regression on the passage-level writing quality reward scores showed *significant positive effect of higher reward model scores for both creativity* ($\beta = 0.26$, SE $= 0.04$, OR $= 1.30$, $p < 0.001$) *and (separately) pragmaticality* ($\beta = 0.28$, SE $= 0.05$, OR $\approx 1.33$, $p < 0.001$). This indicates that existing reward models can be predictive of positive and negative aspects of writing. However, these models output a single passage-level score, making them hard to interpret. In the following section, we explore whether LLMs can provide expression-level creativity judgments.

## 5 CAN LLMS REPLICATE HUMAN CLOSE READING JUDGMENTS OF CREATIVITY?

To understand whether LLMs can be used as expression-level reward models for textual creativity, we evaluate their ability to find both the positive aspects of writing (human-judged perceived novel expressions), as well as the negative ones (non-pragmatic expressions). We formulate a close reading task similar to the annotation setting using our dataset containing passages $\mathcal{P}$, as well as a set of

ground-truth perceived novel expressions (or ground-truth non-pragmatic expressions, respectively) $\mathcal{N}$ provided by 3 annotators. For prompts and hyperparameters, see Appendix C.

**CLOSE READING TASK DEFINITION**   Given a passage, the model needs to extract perceived novel (or non-pragmatic) expressions $\hat{\mathcal{N}}$. We consider expression $\hat{n} \in \hat{\mathcal{N}}$ to be an approximate match with expression $n \in \mathcal{N}$ if one is a subset of the other or their normalized indel similarity (Bachmann, 2025) is $\geq 90\%$. We compute the F1 score as follows: an expression $\hat{n}$ is a true positive if it has an approximate match with some expression in $\mathcal{N}$, and a false positive otherwise. In addition, all $n$ that do not have a match in $\hat{\mathcal{N}}$ are considered as false negatives. Note that achieving a perfect F1 score on this task would mean perfectly modeling every annotator's (subjective) rating. Hence, our goal is not to reach the maximum possible F1 but rather compare the models and check if there is any useful signal in their judgments.

**LLM-AS-A-JUDGE (LLM-J) SETUP**   For reasoning model evaluation, we test 5 frontier models: GPT-5, Claude 4.1, Claude 4.5, Gemini 2.5 Pro, and Gemini 3 Pro. In the zero-shot setting, we provide a prompt analogous to the instructions we gave to the human annotators. In the few-shot setting, we additionally provide examples from 3 passages set aside for few-shot prompting ($\approx$ 15 novel expression and non-pragmatic expression examples) chosen randomly among passages with a median number of novel or non-pragmatic annotations. For finetuned model evaluation, we finetune a set of smaller open-source models (OLMo2-Instruct 7B, Qwen3 8B (Yang et al., 2025), Llama-3.1-Instruct 8B (Meta, 2024)) using LoRA (Hu et al., 2022), as well as GPT-4.1 and Gemini-2.5-Pro using API-based finetuning on 60% of our dataset (including the 3 passages used for few-shot examples), using the rest for evaluation. An empirical random baseline is also provided: for each passage, we extracted $n$ expressions of length $m$ words, where $n$ and $m$ are sampled from negative binomial distributions, with parameters estimated from human annotations using the method of moments.

**RESULTS**   Figure 4 shows F1 scores with 95% confidence intervals for few-shot and finetuned models. There is little difference among models in each task, as well as little improvement from few-shot prompting. The non-pragmatic expression identification task appears to be significantly harder, with F1 scores below 20, whereas model performance on the novel expression task was above 40, cautioning against using LLM-as-a-Judge for identification of writing issues (non-pragmatic expressions). This aligns with our finding earlier that LLMs tend to produce non-pragmatic expressions at high $n$-gram novelty level. All finetuned models lag behind large reasoning models, suggesting that the close reading task is very difficult even with task-specific adaptation. Overall, reasoning models demonstrate impressive performance compared to the empirical random baseline on both tasks. For novelty, random F1 score is 9.6 (95% CI [5.3, 14.0]), which is roughly 4 times lower than reasoning model scores (few-shot GPT-5 Reasoning model F1 $\approx$ 41.3). Similarly, for pragmaticality, the random F1 score is 2.3 (95% CI: [0.0, 4.9]), outperformed by reasoning models by roughly 6 times (few-shot GPT-5 Reasoning model F1 $\approx$ 13.5).

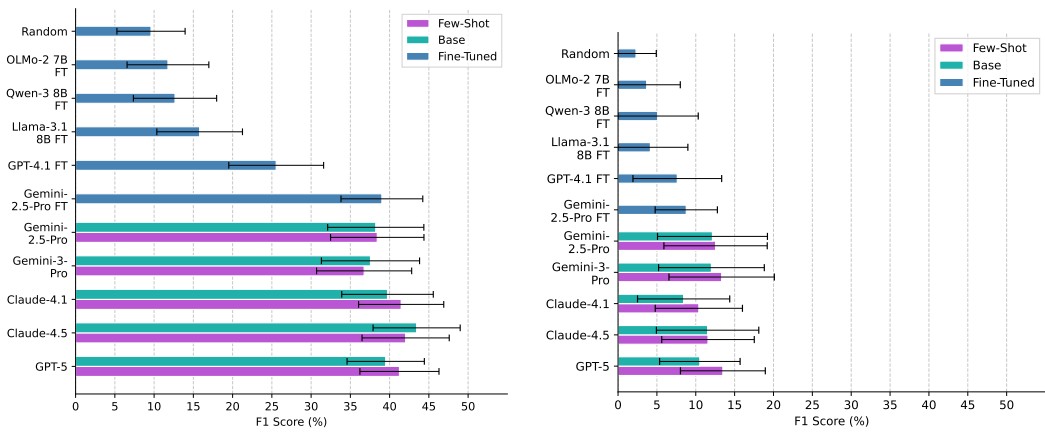

(a) Perceived Novel expression identification

(b) Non-pragmatic expression identification

Figure 4: Comparison of finetuned, zero-shot and few-shot model performance across (a) perceived novel and (b) non-pragmatic expression identification tasks (F1 scores with 95% CIs).

**ALIGNMENT WITH EXPERT PREFERENCES**   We use one of the best performing models on our close reading tasks, few-shot GPT-5, to explore how LLM-as-a-Judge (LLM-J) scores align with expert preferences. Given a dataset of preferences (passageA, passageB, $\mathbb{1}$[A preferred over B]), we obtain the number of novel ($\text{Nov}_A$, $\text{Nov}_B$) and non-pragmatic ($\text{Prag}_A$, $\text{Prag}_B$) expressions identified by the LLM-J for each passage. We normalize each score by the number of words[9] in the passage and take the difference between them to obtain $\Delta\text{Nov}_{AB} = \frac{\text{Nov}_A}{\#\text{ words in }A} - \frac{\text{Nov}_B}{\#\text{ words in }B}$, $\Delta\text{Prag}_{AB} = \frac{\text{Prag}_A}{\#\text{ words in }A} - \frac{\text{Prag}_B}{\#\text{ words in }B}$. A hierarchical logistic regression model is fit on the differences between the scores for every pair of passages predicting expert preference for that pair. We use the Style Mimic dataset containing preferences of three expert writer annotators comparing MFA authors' imitations of famous authors against LLM-generated imitations (Chakrabarty et al., 2025a). Random intercepts were included for each annotator, seed famous author, and for MFA authors nested within seed authors (c.f. Appendix C.6). The model showed that *LLM-J novelty score differences were a significant positive predictor of expert preference* ($\beta_{\Delta\text{Nov}_{AB}} = 0.63$, SE= 0.26, OR = 1.88, $p = 0.014$), while pragmaticality scores were not predictive ($\beta_{\Delta\text{Prag}_{AB}} = -0.05$, $p = 0.832$), as expected due to weaker LLM-as-a-Judge performance on this task.

**ALIGNMENT WITH CROWD PREFERENCES**   We explore whether a similar association is present in the dataset of crowd-sourced writing preferences released by Chakrabarty et al. (2025a) obtained from LMArena Chiang et al. (2024). Each pair contains text generated from 2 language models (model A and model B). Data was sampled such that both model A and model B are in the top 15 most popular models, and so that it contains the same number of comparisons as the Style Mimic data (450). Similarly, we fit a logistic regression model adding random intercepts for the model A and model B. We find that novelty score differences were marginally predictive of the crowd preference ($\beta_{\Delta\text{Nov}_{AB}} = 0.21$, SE= 0.11, OR = 1.24, $p = 0.054$), and pragmaticality scores had a significant negative coefficient ($\beta_{\Delta\text{Prag}_{AB}} = -0.26$, SE= 0.11, OR = 0.77, $p = 0.020$). Together with Style Mimic findings, this indicates that *while LLM-Judge novelty scores align with both expert and crowd preferences, pragmaticality scores only align with crowd preferences*. This may indicate a misalignment between expert and crowd preferences stemming from (mis-)identifying non-pragmatic expressions. Figure 5 visualizes the difference of the LLM-J score relationship with expert vs. crowd preferences.

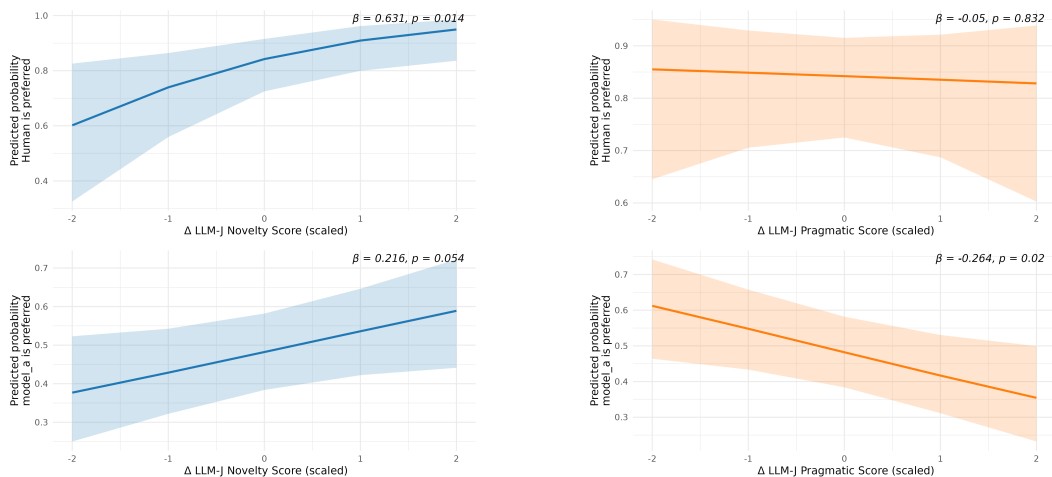

Figure 5: Comparison of how predictive LLM-J novelty ($\Delta\text{Nov}_{AB}$) and pragmaticality ($\Delta\text{Prag}_{AB}$) score differences are of expert vs. crowd preferences, bands indicate 95% confidence intervals.

**COMPARISON WITH CREATIVITY INDEX**   We investigate how well LLM-J novelty scores align with expert preferences compared to CREATIVITY INDEX scores. We compute the CREATIVITY INDEX for each paragraph and use the difference between each pair as a predictor for preference (similarly to the $\Delta\text{Nov}_{AB}$, $\Delta\text{Prag}_{AB}$ scores). While CREATIVITY INDEX was predictive of expert preferences ($\beta_{\Delta\text{CI}_{AB}} = 0.51$, SE= 0.24, OR = 1.66, $p = 0.038$), LLM-J novelty scores had a

---

[9]Not the number of atomic expressions, since LLM-J is allowed to choose expressions regardless of punctuation to mirror the creative highlight setting in the annotation.

slightly stronger effect ($\beta_{\Delta\text{Nov}_{AB}} = 0.63$, $p = 0.014$). In a model including both predictors, LLM-J novelty scores showed a trend-level effect ($\beta_{\Delta\text{Nov}_{AB}} = 0.48$, $p = 0.010$), while CREATIVITY INDEX did not contribute significantly ($\beta_{\Delta\text{CI}_{AB}} = 0.26$, $p = 0.358$). Likelihood ratio tests confirmed that adding CREATIVITY INDEX as a predictor to a model containing novelty scores did not improve model fit ($p = 0.35$), whereas adding novelty scores to a model containing CREATIVITY INDEX showed a trend toward improved fit ($p = 0.096$). Model comparison using AIC (Akaike Information Criterion Akaike (1974), a measure used for statistical model selection) also favored novelty: novelty-only had the lowest AIC ($471.16$) compared with the CREATIVITY INDEX-only model ($473.07$) and the joint model ($472.30$), indicating that LLM-J novelty scores provide the most parsimonious explanation of expert preference.

## 6  RELATED WORK

McCoy et al. (2023) investigates LLM copying from training data, finding that GPT-2 tends to copy large chunks of training corpora and has issues with coherence at high $n$-gram novelty, aligning with some of our findings. Recent advances in training data exploration (Elazar et al., 2024; Merrill et al., 2024; Liu et al., 2024; Xu et al., 2025) have enabled textual creativity metrics based on $n$-gram novelty with respect to trillion-token sized corpora (Lu et al., 2025). Our study cautions against using $n$-gram novelty alone as a metric for creativity based on the standard definition of creativity in psychology requiring the text not only to be novel but also pragmatic. In addition, we make a distinction between human-judged and $n$-gram novelty.

There have been many efforts in quantifying issues in AI writing. Chakrabarty et al. (2025b) consulted with writing experts to create a taxonomy of idiosyncrasies in AI writing. They further collected span level edits from expert writers following the proposed taxonomy to improve AI writing in an automated pipeline. More recently Shaib et al. (2024) collected annotations from experts in terms of what qualifies as "slop" in both human and AI text and found that it correlates with latent dimensions such as coherence and relevance. In a concurrent study Russell et al. (2025) found that people who use AI to write are better detectors of AI writing. Such users typically rely on specific lexical clues ("AI vocabulary"), as well as more complex phenomena within the text such as formality, originality and clarity to discriminate AI from human writing.

Recent work has shown that pervasive LLM use may cause negative societal effects like homogenization (Zhang et al., 2025; Doshi & Hauser, 2024), with RLHF-trained models producing less diverse outputs (Padmakumar & He, 2024). Padmakumar et al. (2025) proposes a novelty metric balancing originality and quality, and corroborates prior finding that LLM text is less novel than human writing (Chakrabarty et al., 2024a). They also find that inference-time methods can boost novelty, though they increase originality at the expense of quality. Unlike prior work, our study investigates the relationship between $n$-gram novelty and human-judged creativity through expert close reading annotations. We demonstrate that high $n$-gram novelty often correlates with reduced pragmaticality in LLM outputs, suggesting that optimizing for novelty alone may not lead to genuinely creative text.

## 7  CONCLUSION

We propose an operationalization of textual creativity beyond $n$-gram novelty. Rooted in the standard definition of creativity, our definition requires assessing both novelty and appropriateness (sensicality and pragmaticality) of text. We conducted a *close reading* study of human and AI-generated text collecting annotations from professional writers ($n = 26$), obtaining a dataset of $8618$ annotated expressions and $274$ creative expression highlights with justifications. Our analysis reveals that as open-source LLMs generate more novel text, they tend to generate less pragmatic expressions, and that $\approx 91\%$ of top-quartile $n$-gram novel expressions are not judged as creative, cautioning against the use of $n$-gram novelty metric alone for creativity evaluation. We confirm similar trends in an exploratory study with frontier models, as well as identify their lower propensity to produce creative expressions compared to humans. We show that reasoning models can replicate some of the human judgments on creativity, aligning with expert preferences in an out-of-distribution dataset, more so than an n-gram based CREATIVITY INDEX.

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

## A DATA

### A.1 SAMPLING EXPRESSIONS FOR ANNOTATION

To select pre-highlighted expressions, a passage was divided into atomic expressions via splitting by punctuation and heuristic rules. Then, we computed % of novel $n$-grams for each expression as follows:

- $E$: the expression
- $\mathcal{G}_n(E)$: all $n$-grams of length $n$ in $T$

- $\mathcal{D}$: reference set of $n$-grams

- $\mathbf{1}\{\cdot\}$: indicator function

$$n^* = \min\Big\{ n \geq 1 : \exists g \in \mathcal{G}_n(E) \text{ with } g \notin \mathcal{D} \Big\}$$

$$\mathrm{NovelPct}(n^*) = \frac{\sum_{g \in \mathcal{G}_{n^*}(E)} \mathbf{1}\{g \notin \mathcal{D}\}}{|\mathcal{G}_{n^*}(E)|}$$

That is, we compute the percentage of novel $n$-grams for $n$ s.t. there is at least one novel $n$-gram. Among the various metrics we tested, this was the most interpretable and had the best balance between correlation with expression length (as we do not want to select only long expressions) and perplexity. We set a threshold for novelty very liberally at $15\%$ which allowed us to get a large number of expressions annotated (roughly $50\%$).

## A.2 SELECTING PASSAGES FOR ANNOTATION

We ensured that passages were not present in the respective OLMo model training corpora by manually checking that large $n$-grams from the passage as well as seemingly rare n-grams (e.g., unique proper nouns) had zero counts in the pre-training corpora. In addition, we had an automatic verification mechanism where a sample of 5 15-grams from the beginning, 5 from middle, and 5 from the end of the text were checked to have a 0 count in the pretraining data.

To sample the 5 passages for the follow-up exploratory study with frontier models, we split the 50 passages from the main study into novelty quintiles according to estimates from the model novel $\sim$ $1 + (1|\mathrm{para\_id}) + (1|\mathrm{annotator})$. We randomly pick one passage per quintile.

## A.3 ANNOTATION INSTRUCTIONS

We provide the annotation instructions in Figure 6.

## A.4 DATASET EXAMPLES

Table 3 shows additional examples of low $n-$gram novelty creative expressions.

## A.5 ANNOTATOR AGREEMENT COMPUTATION

We use `statsmodels` package (Skipper & Josef, 2010) to compute the free-marginal Kappa.

| *Expression* | **Justification** | **PPL** |
|---|---|---|
| *disappearing from one room and reappearing in another* | As in the previous clause, this metaphor defamiliarizes a mundane experience comparing it to the magical, heightening the surreality/drama of the narrator watching their daughter. | -1.6 |
| *you're the ugliest woman I've ever seen* | This sentence marks a surprising turn from what has been an innocuous description of a date up to this point. | -1.5 |
| *the days of the week whisk by like panties* | Inventive simile; surprising and humorous, central to the thematic conceit of the passage. Strong voice. | -1.4 |
| *lay across the roof of the house like the severed hand of a giant* | Describing a felled pine bough as similar to "the severed hand of a giant" is delightfully fresh and original. | -1.4 |

Table 3: Examples of expressions judged creative with low $n$-gram novelty (log-standardized perplexity), along with the expert writers' justifications.

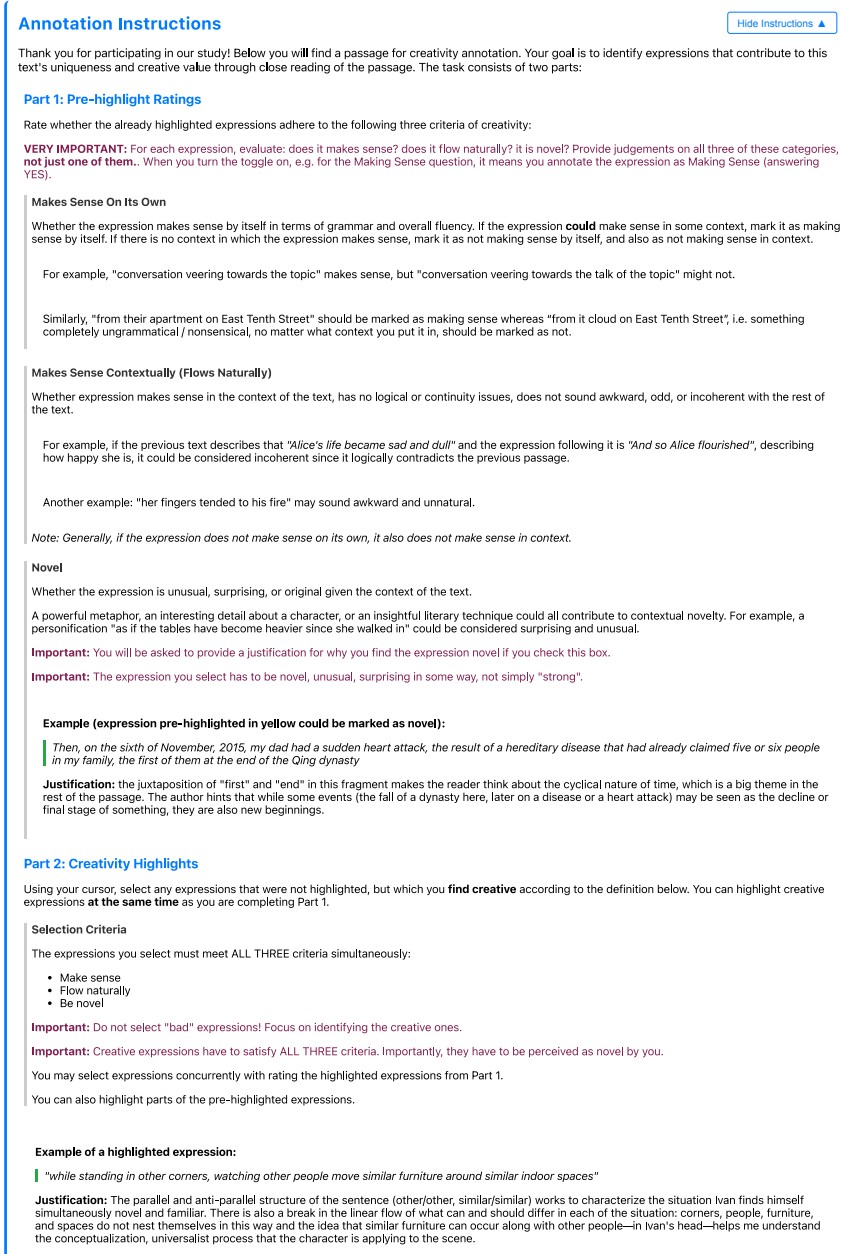

Figure 6: Annotation Instructions

# B  LINEAR MODELS

All linear models herein were estimated with the `lme4` R package (Bates et al., 2015).

## B.1  DATA PRE-PROCESSING

Due to instability of the infinigram API, we intially were not able to detect two passages as part of the OLMo pretraining data (IDs 14520 and 7644). We therefore removed those passages at the pre-processing stage for the linear models, which did not affect the outcomes of the research questions. In addition, we filter out 37 unique expressions that received varying perplexity estimates from the infinigram API due to being computed at different times.

## B.2 N-GRAM NOVELTY AND CREATIVITY

We incorporate creativity highlights in the following manner: we add all highlighted expressions and their perplexity scores to the pre-highlighted ratings data, exclude expressions that were subsets, supersets or over 90% similar by normalized indel similarity Bachmann (2025). Given that highlights allowed the annotators to select any expression, we consider expressions that were not highlighted (and were not in the pre-highlighted set) as non-novel by any annotator that did not highlight it. As a result, our dataset contains an enlarged number of $15,982$ annotations.

|  | creative ~ ppl_log_std + gen_source + (1 \| annotator) + (1 \| seed_passage_id) |
| --- | --- |
| (Intercept) | $-3.29\ [-3.64; -2.94]^*$ |
| ppl_log_std | $0.67\ [0.59; 0.76]^*$ |
| gen_sourceolmo1 | $-0.63\ [-0.87; -0.38]^*$ |
| gen_sourceolmo2 | $-0.61\ [-0.85; -0.37]^*$ |
| $R_m^2$ (theoretical) | 0.12 |
| $R_c^2$ (theoretical) | 0.30 |
| AIC | 5066.55 |
| BIC | 5112.53 |
| Log Likelihood | $-2527.28$ |
| Num. obs. | 15725 |
| Num. groups: seed_passage_id | 50 |
| Num. groups: annotator | 26 |
| Var: seed_passage_id (Intercept) | 0.36 |
| Var: annotator (Intercept) | 0.53 |

$^*$ 0 outside the confidence interval.

Table 4: Mixed-effects logistic regression predicting creativity from standardized PPL

## B.3 N-GRAM NOVELTY AND PRAGMATICALITY

To focus on pragmaticality, we exclude expressions rated as non-sensical (so we know they do not make sense in the context, not just because they are ill-formed). The full specification of the model is available in Table 5.

For frontier model study, we did not exclude these expressions since the sample size of non-pragmatic expressions was already very small (12 and 43 for Claude and GPT-5). In addition, since our research question concerns a head-to-head comparison of humans and frontier LLMs, we exclude passages generated by OLMo and OLMo-2. The model is available in Table 6.

We tested the following null hypotheses regarding the effect of perplexity (`ppl_log_std`) on pragmatic judgments within each generator:

**Human (reference group):**     $H_0 : \beta_{\text{ppl\_log\_std}} = 0$

**LLM:**     $H_0 : \beta_{\text{ppl\_log\_std}} + \beta_{\text{ppl\_log\_std:gen\_sourceLLM}} = 0$

The tests were carried out with the function `linearHypothesis()` from the `car` package Fox & Weisberg (2019), applied to the fitted `glmer` model.

## B.4 CREATIVITY AND GENERATION SOURCE

To compare creativity along all generation sources, we add the frontier model annotations to the original data, adding the highlights in the same manner as above. The model is available in Table 7.

## B.5 WRITING QUALITY REWARD MODELS AND CREATIVITY

The model for creativity is available in Table 8, and for pragmaticality in Table 9.

| | pragmatic ˜ ppl_log_std * gen_source + (1 \| annotator) + (1 \| seed_passage_id) |
|---|---|
| (Intercept) | $3.93\ [3.31; 4.55]^{*}$ |
| ppl_log_std | $0.01\ [-0.16; 0.18]$ |
| gen_sourceolmo1 | $-1.50\ [-1.81; -1.18]^{*}$ |
| gen_sourceolmo2 | $0.43\ [0.07; 0.78]^{*}$ |
| ppl_log_std:gen_sourceolmo1 | $-0.18\ [-0.42; 0.05]$ |
| ppl_log_std:gen_sourceolmo2 | $-0.49\ [-0.80; -0.18]^{*}$ |
| $R_m^2$ (theoretical) | 0.10 |
| $R_c^2$ (theoretical) | 0.47 |
| AIC | 2752.73 |
| BIC | 2808.07 |
| Log Likelihood | $-1368.37$ |
| Num. obs. | 7455 |
| Num. groups: seed_passage_id | 50 |
| Num. groups: annotator | 26 |
| Var: seed_passage_id (Intercept) | 0.31 |
| Var: annotator (Intercept) | 2.00 |

$^{*}$ 0 outside the confidence interval.

Table 5: Mixed-effects logistic regression predicting pragmatism from standardized PPL and gen. source

| | pragmatic ˜ ppl_log_std * gen_source + (1 \| annotator) + (1 \| seed_passage_id) |
|---|---|
| (Intercept) | $2.70\ [1.67; 3.73]^{*}$ |
| ppl_log_std | $0.13\ [-0.24; 0.49]$ |
| gen_sourceclaude | $1.47\ [-0.09; 3.04]$ |
| gen_sourcegpt5 | $0.04\ [-1.46; 1.53]$ |
| ppl_log_std:gen_sourceclaude | $-0.35\ [-1.02; 0.32]$ |
| ppl_log_std:gen_sourcegpt5 | $-0.30\ [-0.80; 0.20]$ |
| $R_m^2$ (theoretical) | 0.10 |
| $R_c^2$ (theoretical) | 0.41 |
| AIC | 613.98 |
| BIC | 656.38 |
| Log Likelihood | $-298.99$ |
| Num. obs. | 1481 |
| Num. groups: annotator | 12 |
| Num. groups: seed_passage_id | 5 |
| Var: annotator (Intercept) | 1.62 |
| Var: seed_passage_id (Intercept) | 0.09 |

$^{*}$ 0 outside the confidence interval.

Table 6: Mixed-effects logistic regression predicting creativity from standardized PPL (comparing humans and frontier models)

|  | creative ~ gen_source + (1 \| annotator) + (1 \| seed_passage_id) |
|---|---|
| (Intercept) | $-3.08\ [-3.42; -2.74]^*$ |
| gen_sourceclaude | $-0.65\ [-1.12; -0.18]^*$ |
| gen_sourcegpt5 | $-0.67\ [-1.16; -0.19]^*$ |
| gen_sourceolmo1 | $-0.69\ [-0.93; -0.45]^*$ |
| gen_sourceolmo2 | $-0.53\ [-0.76; -0.30]^*$ |
| $R_m^2$ (theoretical) | 0.02 |
| $R_c^2$ (theoretical) | 0.22 |
| AIC | 5898.85 |
| BIC | 5953.43 |
| Log Likelihood | $-2942.42$ |
| Num. obs. | 18004 |
| Num. groups: seed_passage_id | 50 |
| Num. groups: annotator | 26 |
| Var: seed_passage_id (Intercept) | 0.33 |
| Var: annotator (Intercept) | 0.51 |

$^*$ 0 outside the confidence interval.

Table 7: Mixed-effects logistic regression predicting creativity from generation source (comparing humans and all models)

|  | creative ~ WQRM_score + (1 \| annotator) + (1 \| seed_passage_id) |
|---|---|
| (Intercept) | $-5.36\ [-6.01; -4.71]^*$ |
| WQRM_score | $0.26\ [0.19; 0.33]^*$ |
| $R_m^2$ (theoretical) | 0.02 |
| $R_c^2$ (theoretical) | 0.22 |
| AIC | 5339.09 |
| BIC | 5369.74 |
| Log Likelihood | $-2665.54$ |
| Num. obs. | 15725 |
| Num. groups: seed_passage_id | 50 |
| Num. groups: annotator | 26 |
| Var: seed_passage_id (Intercept) | 0.34 |
| Var: annotator (Intercept) | 0.49 |

$^*$ 0 outside the confidence interval.

Table 8: Mixed-effects logistic regression predicting creativity from passage-level WQRM scores

| | pragmatic ~ WQRM_score + (1 \| annotator) + (1 \| seed_passage_id) |
|---|---|
| (Intercept) | 1.48 $[0.56; 2.41]^*$ |
| WQRM_score | 0.28 $[0.19; 0.38]^*$ |
| $R^2_m$ (theoretical) | 0.02 |
| $R^2_c$ (theoretical) | 0.45 |
| AIC | 2816.08 |
| BIC | 2843.75 |
| Log Likelihood | $-1404.04$ |
| Num. obs. | 7455 |
| Num. groups: seed_passage_id | 50 |
| Num. groups: annotator | 26 |
| Var: seed_passage_id (Intercept) | 0.74 |
| Var: annotator (Intercept) | 1.85 |

$^*$ 0 outside the confidence interval.

Table 9: Mixed-effects logistic regression predicting pragmaticality from passage-level WQRM scores

### B.6 AI-LIKELIHOOD SCORES

The model for creativity is available in Table 10, and for pragmaticality in Table 11.

| | creative ~ std_log_ai_likelihood + (1 \| annotator) + (1 \| seed_passage_id) |
|---|---|
| (Intercept) | $-3.08 \; [-3.43; -2.74]^*$ |
| std_log_ai_likelihood | $0.05 \; [-0.16; 0.27]$ |
| $R^2_m$ (theoretical) | 0.00 |
| $R^2_c$ (theoretical) | 0.20 |
| AIC | 3127.29 |
| BIC | 3154.94 |
| Log Likelihood | $-1559.64$ |
| Num. obs. | 7430 |
| Num. groups: seed_passage_id | 49 |
| Num. groups: annotator | 26 |
| Var: seed_passage_id (Intercept) | 0.31 |
| Var: annotator (Intercept) | 0.52 |

$^*$ 0 outside the confidence interval.

Table 10: Mixed-effects logistic regression predicting creativity from passage-level AI likelihood scores

## C    LLM-JUDGE EVALUATION DETAILS, PROMPTS AND HYPERPARAMETERS

### C.1    CONFIDENCE INTERVAL COMPUTATION

All 95% CIs computed with with `confidenceinterval` package (Gildenblat, 2023) using the Takahashi et al. (2022) method.

|  | pragmatic ~ std_log_ai_likelihood + (1 \| annotator) + (1 \| seed_passage_id) |
| --- | --- |
| (Intercept) | $3.85 \; [3.27; 4.43]^{*}$ |
| std_log_ai_likelihood | $-0.25 \; [-0.53; 0.04]$ |
| $R_m^2$ (theoretical) | 0.01 |
| $R_c^2$ (theoretical) | 0.37 |
| AIC | 1042.47 |
| BIC | 1067.07 |
| Log Likelihood | $-517.24$ |
| Num. obs. | 3466 |
| Num. groups: seed_passage_id | 49 |
| Num. groups: annotator | 26 |
| Var: seed_passage_id (Intercept) | 0.37 |
| Var: annotator (Intercept) | 1.48 |

$^{*}$ 0 outside the confidence interval.

Table 11: Mixed-effects logistic regression predicting pragmaticality from passage-level WQRM scores

## C.2 OLMo VERSIONS, PROMPTS AND GENERATION HYPERPARAMTERS

To generate LLM passages for annotation, we use the following simple prompts for OLMo (`OLMo-7B-0724-Instruct`) and OLMo-2 (`OLMo-2-0325-32B-Instruct`) in Table 12.

| Row | Prompt |
| --- | --- |
| Summary Prompt | `{story}\n\nSummarize in one sentence.` |
| Passage Generation Prompt | `{summary}\n\nWrite a fragment of a story around {n_words} words long in the style of {author} based on the summary above.` |

Table 12: Prompts for summary and passage generation.

We sample with `temperature` $= 1$ and truncate the output by paragraphs if they were too long, regenerating if they were too short (not within 10% of the original passage). For summaries, we manually ensured the overall topic was consistent with the passage (in 2-3 rare cases, the summaries were completely off-topic). For stories, we manually examined the outputs and regenerated the story in case of an output that did not at all adhere to the prompt (for example, sometimes OLMo model would generate a a play script instead of a passage, and once the model generated a story fully in Spanish).

We use the same generation prompt for Claude-4.1 and GPT-5 in the exploratory frontier model study, and use default API inference hyperparameters. We used the same summaries as for the OLMo models to exclude summary content effects.

## C.3 LLM-AS-A-JUDGE VERSIONS AND GENERATION HYPERPARAMTERS

- **GPT-5 Hyperparameters:**
  - Model version: `gpt-5-2025-08-07`
  - verbosity: high
  - reasoning effort: high
- **Claude-4.1 Hyperparameters:**
  - Model version: `claude-opus-4-1-20250805`

      – temperature = 1
- **Gemini 2.5 Pro Hyperparameters:** Default hyperparameters

## C.4 PROMPTS FOR MODEL EVALUATION

We based our prompts very closely on annotator instructions (see Figure 6). Figure 7 shows the prompts used for zero-shot and few-shot model evaluation.

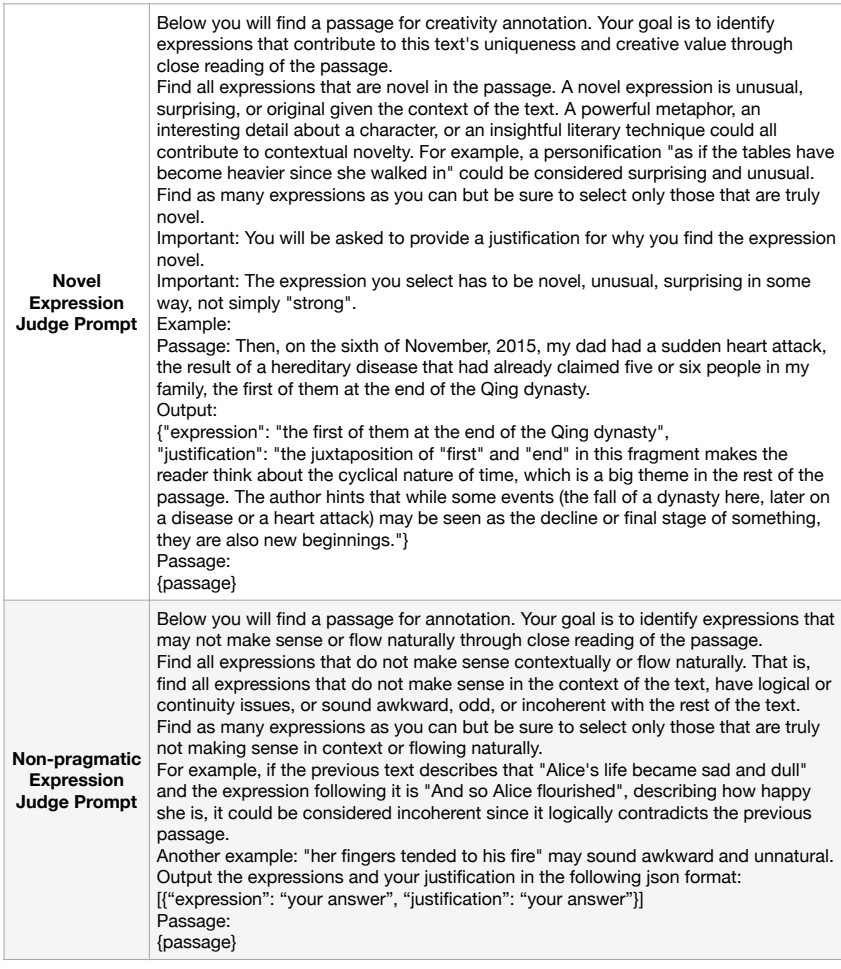

Figure 7: Prompts used for few-shot and zero-shot model evaluation.

## C.5 FINETUNING HYPERPARAMETERS

Below are the hyperparameters used for model fine-tuning:

- **GPT-4.1**
  - Model version gpt-4.1-2025-04-14
  - Fine-tuning type: full parameter (API-based)
  - Batch size: 2
  - Learning rate multiplier: 2
  - Epochs: 3
  - Seed: 42
- **OLMo-2-7B**

- **–** Model version: `OLMo-2-1124-7B-Instruct`
- **–** Fine-tuning type: LoRA
  - **∗** LoRA rank: 8
  - **∗** LoRA alpha: 8
  - **∗** lora dropout 0.2
- **–** per device batch size: 1
- **–** gradient accumulation steps: 2
- **–** learning rate 2e-4
- **–** scheduler type linear
- **–** warmup ratio 0.03
- **–** weight decay 0.01
- **–** epochs 3
- **–** max seq length 4096

- **Qwen3-8B**
  - **–** Model version: `Qwen3-8B`
  - **–** Fine-tuning parameters: same as above

- **LLama-3.1**
  - **–** Model version: `Llama-3.1-8B-Instruct`
  - **–** Fine-tuning parameters: same as above

All open model fine-tuning was ran on 2 A100 40GB NVIDIA GPUs. We used the Open-Instruct codebase for fine-tuning script Lambert et al. (2024).

## C.6 LINEAR MODELS FOR EXPERT AND CROWD PREFERENCE ALIGNMENT

The linear model fit to esimtate alignment between LLM-J scores and expert references in presented in Table 13. For crowd preferences, see Table 14. AIC comparison and model specifications for Creativity Index comparisons can be found in Table 15.

| | preference_bin ~ scaled_delta_nov_score + scaled_delta_prag_score + (1 \| annotator) + (1 \| seed_author / mfa_author) |
|---|---|
| (Intercept) | $1.67\ [0.97; 2.38]^*$ |
| scaled_delta_nov_score | $0.63\ [0.13; 1.13]^*$ |
| scaled_delta_prag_score | $-0.05\ [-0.52; 0.41]$ |
| $R_m^2$ (theoretical) | 0.04 |
| $R_c^2$ (theoretical) | 0.62 |
| AIC | 473.11 |
| BIC | 497.77 |
| Log Likelihood | $-230.56$ |
| Num. obs. | 450 |
| Num. groups: mfa_author:seed_author | 150 |
| Num. groups: seed_author | 50 |
| Num. groups: annotator | 21 |
| Var: mfa_author:seed_author (Intercept) | 3.49 |
| Var: seed_author (Intercept) | 0.92 |
| Var: annotator (Intercept) | 0.51 |

$^*$ 0 outside the confidence interval.

Table 13: Mixed-effects logistic regression predicting expert style preference from novelty and pragmaticity score differences

| | preference_bin ˜ scaled_delta_nov_score + scaled_delta_prag_score + (1 \| model_a) + (1 \| model_b) |
|---|---|
| (Intercept) | $-0.07\ [-0.47; 0.33]$ |
| scaled_delta_nov_score | $0.22\ [-0.00; 0.44]$ |
| scaled_delta_prag_score | $-0.26\ [-0.49; -0.04]^*$ |
| $R^2_m$ (theoretical) | 0.02 |
| $R^2_c$ (theoretical) | 0.14 |
| AIC | 609.06 |
| BIC | 629.61 |
| Log Likelihood | $-299.53$ |
| Num. obs. | 450 |
| Num. groups: model_a | 15 |
| Num. groups: model_b | 15 |
| Var: model_a (Intercept) | 0.10 |
| Var: model_b (Intercept) | 0.36 |

$^*$ 0 outside the confidence interval.

Table 14: Mixed-effects logistic regression predicting crowd preference from novelty and pragmaticality score differences

| | (1) Nov. only | (2) CR-index only | (3) Nov. + CR-index |
|---|---|---|---|
| (Intercept) | $1.67\ [0.97; 2.38]^*$ | $1.71\ [1.00; 2.43]^*$ | $1.69\ [0.98; 2.40]^*$ |
| scaled_delta_nov_score | $0.62\ [0.13; 1.11]^*$ | | $0.48\ [-0.09; 1.05]$ |
| scaled_delta_crindex_score | | $0.51\ [0.03; 0.99]^*$ | $0.26\ [-0.29; 0.80]$ |
| $R^2_m$ (theoretical) | 0.04 | 0.03 | 0.05 |
| $R^2_c$ (theoretical) | 0.62 | 0.62 | 0.62 |
| AIC | 471.16 | 473.07 | 472.30 |
| BIC | 491.70 | 493.61 | 496.95 |
| Log Likelihood | $-230.58$ | $-231.53$ | $-230.15$ |
| Num. obs. | 450 | 450 | 450 |
| Num. groups: mfa_author:seed_author | 150 | 150 | 150 |
| Num. groups: seed_author | 50 | 50 | 50 |
| Num. groups: annotator | 21 | 21 | 21 |
| Var: mfa_author:seed_author (Intercept) | 3.51 | 3.82 | 3.53 |
| Var: seed_author (Intercept) | 0.90 | 0.80 | 0.89 |
| Var: annotator (Intercept) | 0.51 | 0.52 | 0.53 |

$^*$ Null hypothesis value outside the confidence interval.

Table 15: Mixed-effects logistic regression predicting expert style preference: novelty only, CR-index only, and combined. All models include random intercepts for annotator and seed_author/mfa_author.

## D ADDITIONAL FIGURES

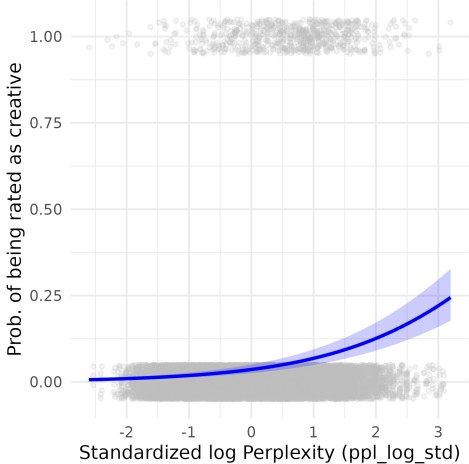

Figure 8: Effect of $n$-gram novelty (log-standardized perplexity) on probability of the expression being rated as novel.

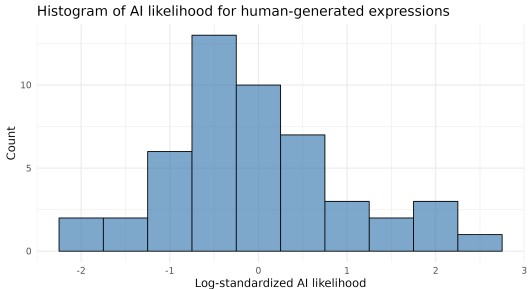

Figure 9: passage-level log AI-likelihood distribution for human-written expressions.

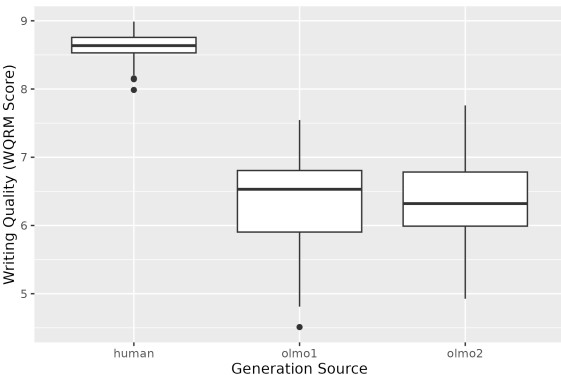

Figure 10: Writing quality reward model distribution by generation source.

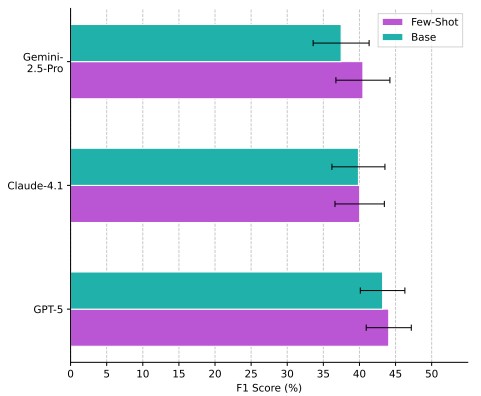

(a) **Novel Expression Identification:** few-shot and zero-shot results.

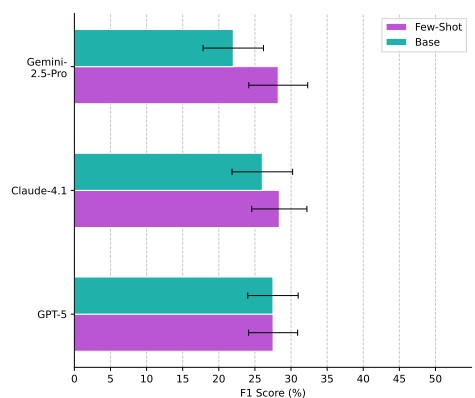

(b) **Non-Pragmatic Expression Identification:** few-shot and zero-shot results.

Figure 11: Comparison of zero-shot and few-shot model performance across (a) novel and (b) non-pragmatic expression identification tasks on the full test set of 97% of the data excluding 3 few-shot passages only (F1 scores with 95% CIs computed with `confidenceinterval` package (Gildenblat, 2023) using the Takahashi et al. (2022) method.

