# OpenReview forum: "Death of the Novel(ty): Beyond N-Gram Novelty as a Metric for Textual Creativity"
_ICLR.cc/2026/Conference — ICLR 2026 Poster_

### Official Review · Reviewer_Z5SQ · 2025-10-31

**Soundness:** 3
**Presentation:** 3
**Contribution:** 2
**Rating:** 4
**Confidence:** 3

**Summary:**

This paper challenges the use of n-gram novelty as a sufficient metric for evaluating textual creativity in language models. The authors argue that true creativity requires both novelty (how original the text is) and appropriateness (how sensical and pragmatic it is), based on established creativity theory. Through a study with 26 expert writers annotating 7542 expressions from human and AI-generated text, they find that 91% of top-quartile expressions by n-gram novelty are not judged as creative, and that higher n-gram novelty in open-source LLMs in facts correlates with lower pragmaticality. Furthermore, they found that LLMs are less likely to produce creative expressions than humans and even struggle to identify creative expressions (a positive aspect of writing) and non-pragmatic ones (a negative aspect).

**Strengths:**

1. The paper is well motivated, well written, and addresses an important research question on measuring creativity.
2. The authors conduct high-quality expert annotations, recruited 26 professional writers from top MFA programs with rigorous training, achieving good inter-rater agreement (κ_free = 0.78 for novelty, 0.72 for pragmaticality)
3. The finding is somewhat surprising: 91% of top-quartile expressions by n-gram novelty are not judged as creative by experts, highlighting the risk of over-relying on n-gram based metrics as a proxy for creativity.

**Weaknesses:**

1. The scale of the study is relatively small. For example, the human text samples include only 50 passages of roughly 400 words each.

2. The domain is limited to creative fiction from The New Yorker, findings may or may not generalizes to other domains such as technical writing, journalism, or poetry, where measuring creativity is also very important.

3. The authors rely on Infinigram and open-source OLMo/OLMo-2 corpora to compute n-gram novelty. However, the training data of frontier models (e.g., GPT-5, Claude 4.1) are not publicly available, raising the question of whether the observed limitations of n-gram novelty metric comes from the n-gram metric itself or from an imprecise estimation of n-gram novelty due to incomplete corpus coverage.

**Questions:**

What do you think it's a scalable method to measure creativity beyond n-gram metrics and expert annotation?

---

> ### Author Response · Authors · 2025-11-20
> **Response to Reviewer Z5SQ [1/2]**
>
> We thank the reviewer for their assessment of our work as well-motivated, well-written, addressing important questions, and having surprising findings. We hope to address some of the concerns below:
>
> ### 1. Study Scale
> > The scale of the study is relatively small. For example, the human text samples include only 50 passages of roughly 400 words each.
>
> Although we work with 50 human and 50 AI passages, each one yields many fine-grained expert annotations. Specifically:
> - 7,542 expression ratings across sensicality, pragmaticality, and novelty (with written justifications)
> - 226 expert-highlighted creative expressions with written justifications
> - These annotations also come from a pool of 26 professional writers (larger than typical expert-annotation studies, see response to NcHT)
>
> - The large number of observations allows us to obtain statistically significant results across our main claims:
>     - **Inadequacy of n-gram novelty**: Among top-quartile n-gram novel expression ratings, 91% were not judged creative *(Wilson 95% CI: [0.90, 0.92]; n = 3,995 expression ratings)*. As the confidence interval is already very narrow, additional passages would not affect the substantive claim that n-gram novelty has a high false-positive rate.
>     - **Negative correlation between n-gram novelty and pragmaticality**: the regression yields β = -0.48, p < 0.001, indicating a statistically and practically significant effect.
>     - **Human vs. LLM creativity rates**: All models show significantly lower creativity prevalence than humans (all p < 0.03, all 95% CIs < 1; Table 2), note we corrected for multiple comparisons via Dunnett’s method.
>     - **Writing quality reward model correlations**: WQRM scores predict both creativity and pragmaticality (OR 1.27-1.32, p < 0.001).
>     - **LLM-as-a-judge alignment**: LLM-as-a-judge novelty scores predict expert preferences (OR = 1.88, p = 0.014).
>
> - Close reading by professional writers is inherently resource-intensive (2-3 hours per 10 passages). They were tasked with rating multiple dimensions (sensicality, pragmaticality, novelty) while also identifying creative expressions with written justifications. Comparable close-reading expert creativity studies use similar scale. E.g., [1] annotates 48 stories with 10 annotators while our study includes more annotators (26) and more granular expression level-judgements. See more comparisons in Response to NcHT.
>
> To summarize, increasing the number of passages would be very costly and would not meaningfully change our findings. We emphasize that we are releasing the complete dataset to allow the community to validate and build upon our work.
>
> ### 2. Fiction domain
>
> > The domain is limited to creative fiction from The New Yorker, findings may or may not generalizes to other domains such as technical writing, journalism, or poetry, where measuring creativity is also very important.
>
> We focus on the fiction domain because creativity is its main objective, as opposed to journalism or technical writing where factuality is equally (if not more) important. Creativity judgments for poetry have a higher degree of subjectivity than fiction [2,3], requiring more raters with high expertise to achieve a high-powered study. For these reasons, we turned to professionally edited, ‘gold-standard’ fiction from The New Yorker (e.g., authors in our sample include several Nobel prize-winning authors).
>
> Additional domain evaluation (taking into account the limitations outlined above) is an important direction for future work, and we will release our code and data to support such efforts.
>
> ### 3. Use of open-source models and corpora
>
> > The authors rely on Infinigram and open-source OLMo/OLMo-2 corpora to compute n-gram novelty. However, the training data of frontier models (e.g., GPT-5, Claude 4.1) are not publicly available, raising the question of whether the observed limitations of n-gram novelty metric comes from the n-gram metric itself or from an imprecise estimation of n-gram novelty due to incomplete corpus coverage.
>
> We agree that n-gram novelty can only be precisely estimated when the pretraining corpus is known. Importantly:
> - *All core findings* about the limitations of n-gram novelty as a metric (e.g., high false-positive rate) are based *exclusively on OLMo and OLMo-2*, where the full training corpora are *publicly available.*
> - We ensured that none of the human passages appear in these corpora, so novelty estimates for the main analysis are reliable.
> - While we cannot guarantee coverage for close-source models, OLMo corpora are large and diverse, making them a reasonable proxy for exploratory comparisons.
> - The analysis of frontier models is explicitly exploratory. These models are included only to confirm that the general trends observed in open-source models (lower creativity rate, negative correlation with pragmaticality) generalize to close-source models.

---

> > ### Author Response · Authors · 2025-11-20
> > **Response to Reviewer Z5SQ [2/2]**
> >
> > ### Answers to questions
> >
> > > What do you think it's a scalable method to measure creativity beyond n-gram metrics and expert annotation?
> >
> > We agree that scalable evaluation of creativity is an important open problem. As we demonstrate in our work, the current scalable approach based on n-gram novelty is conceptually and practically flawed. Our primary contribution is to rigorously define what a correct metric should measure based on prior work in psychology of creativity.
> >
> > LLM-as-a-Judge (LLM-J) few-shot prompted with our data presents a proof-of-concept that this theoretically-grounded definition can be successfully operationalized. Specifically, we find that LLM-J novelty scores provide a promising signal on unseen datasets:
> > - They are a statistically significant predictor of expert preference (p=0.014) (lines 376-378)
> > - They show alignment with OOD crowd preferences (p=0.054, Appendix D)
> > - They outperform n-gram based Creativity Index (Appendix E) in predicting expert preferences.
> >
> > Further work can explore techniques such as synthetic reasoning traces (to train reasoning models) or RL training based on our expert-annotated spans to improve creativity annotation performance.
> >
> > In addition, we show that Writing Quality Reward Models introduced in prior work are also significantly predictive of creativity judgements (297-305).
> >
> > We thank the reviewer for the productive feedback which will help us improve our paper! If any concerns remain unaddressed or there are any questions, please let us know. We will be uploading a revision of the paper incorporating your feedback shortly, please let us know if there are any more comments!
> >
> > ### References
> >
> > [1] Chakrabarty, Tuhin, et al. "Art or artifice? large language models and the false promise of creativity." Proceedings of the 2024 CHI Conference on Human Factors in Computing Systems. 2024.
> >
> > [2] Kaufman, James C., et al. "A comparison of expert and nonexpert raters using the consensual assessment technique." Creativity Research Journal 20.2 (2008): 171-178.
> >
> > [3] KAUFMAN, J.C., BAER, J. and COLE, J.C. (2009), Expertise, Domains, and the Consensual Assessment Technique. The Journal of Creative Behavior, 43: 223-233. https://doi.org/10.1002/j.2162-6057.2009.tb01316.x

---

### Official Review · Reviewer_NcHT · 2025-11-01

**Soundness:** 3
**Presentation:** 2
**Contribution:** 2
**Rating:** 6
**Confidence:** 3

**Summary:**

This paper declared death of n-gram as a measure of creativity, with detailed study into rare n-gram in both human & AI generated passages and painstakingly annotated dataset. Findings are cool: rare n-grams are overwhelmingly not creative. The author continue to investigate if AI can replace human in such annotations.

**Strengths:**

- This is a well-motivated and complete investigation into whether n-gram can be a good metric.
- The author conducted rigorous annotations on n-gram quality and creativity, with the annotation guideline elaborate and well defined. Annotations of the samples are really well thought and detailed. Statistical analysis are thorough.
- The author investigated various interesting research quesitons that flows well with the ideas, including if LLM can replace human in close reading and if experts are needed to create the best annotations.

**Weaknesses:**

- The scope of the work is clear but limited to New Yorker passages and n=26.
- This result is intuitive but expected, once one understand the context. While the work is solid, its insights are not very novel by iclr standards.
- The paper's writing can benefit from placing some details (e.g. python packages used) in appendix and devote more space to more analysis.
- LLM as a judge's F1 performance is overly low. Some more insights would be helpful. The current statement "it suggests it may be stemming from failure to recognize non-pragmatic expressions" is unconvincing.
- Similarly, finetuned pragmatic or novelty identification models perform poorly, with error bars swing to as bad as x = 0. A further invesitgation on why, or finetuning larger model might be needed.

**Questions:**

Please see weakness.

---

> ### Author Response · Authors · 2025-11-20
> **Response to Reviewer  NcHT [1/n]**
>
> We thank the reviewer for their assessment of our work as well-motivated, complete, rigorous, and interesting, and for their insightful comments. We hope to address some of the concerns below:
>
> ### 1. n=26 expert writers and The New Yorker as a source
> > The scope of the work is clear but limited to New Yorker passages and n=26.
>
> - We focus on the fiction domain because creativity is its main objective (as opposed to e.g. journalism or technical writing where factuality is equally (if not more) important). For these reasons, we turned to professionally edited, ‘gold-standard’ fiction from The New Yorker (e.g., authors in our sample include Nobel prize-winning authors).
>
> - As for the 26 expert annotators, we exceed the typical number of participants in expert creativity studies:
> | Paper / Citation                                                                                                                           | Expert n |
> |--------------------------------------------------------------------------------------------------------------------------------------------|-------|
> |Kaufman, James C., et al. "A comparison of expert and nonexpert raters using the consensual assessment technique." Creativity Research Journal 20.2 (2008): 171-178.                 | 10    |
> | Kaufman, James C., John Baer, and Jason C. Cole. "Expertise, domains, and the consensual assessment technique." The Journal of creative behavior 43.4 (2009): 223-233. | 10    |
> | Hu, Linna, May Boggess, and Mardelle M. Shepley. "Comparing expert, quasi-expert, and novice evaluations of award-winning design products using the consensual assessment technique." Creativity Research Journal 35.4 (2023): 730-748.   | 13    |
> | Chakrabarty, Tuhin, et al. "Art or artifice? large language models and the false promise of creativity." Proceedings of the 2024 CHI Conference on Human Factors in Computing Systems. 2024.                                                   | 10    |
> | Stevenson, Claire E., et al. "Putting GPT-3's Creativity to the (Alternative Uses) Test." ICCC. 2022.                                                 | 2     |
> | Koivisto, Mika, and Simone Grassini. "Best humans still outperform artificial intelligence in a creative divergent thinking task." Nature Scientific reports 13.1 (2023): 13601.             | 6     |
> | Dollinger, S. J., & Shafran, M. (2005). *Note on Consensual Assessment Technique in Creativity Research*. Perceptual and Motor Skills      | 5     |
> | *Measuring AI "Slop" in Text* (Shaib et al., 2025)                                                                                         | 5     |
>
> Note that the overall number of observations far exceeds 26. Specifically we obtain:
>
> - 7,542 expression ratings across sensicality, pragmaticality, and novelty (with written justifications)
> - 226 expert-highlighted creative expressions with written justifications
> - 26 professional writers (larger than typical expert-annotation studies)
>
> The large number of observations allows us to obtain statistically significant results across our main claims (as mentioned in Response to Z5SQ):
> - **Inadequacy of n-gram novelty**: Among top-quartile n-gram novel expression ratings, 91% were not judged creative *(Wilson 95% CI: [0.90, 0.92]; n = 3,995 expression ratings)*. As the confidence interval is already very narrow, additional passages would not affect the substantive claim that n-gram novelty has a high false-positive rate.
> - **Negative correlation between n-gram novelty and pragmaticality**: the regression yields β = -0.48, p < 0.001, indicating a statistically and practically significant effect.
> - **Human vs. LLM creativity rates**: All models show significantly lower creativity prevalence than humans (all p < 0.03, all 95% CIs < 1; Table 2), note we corrected for multiple comparisons via Dunnett’s method.
> - **Writing quality reward model correlations**: WQRM scores predict both creativity and pragmaticality (OR 1.27-1.32, p < 0.001).
> - **LLM-as-a-judge alignment**: LLM-as-a-judge novelty scores predict expert preferences (OR = 1.88, p = 0.014).
>
> To summarize, the current number of annotators exceeds prior work, increasing the number of annotators would be very costly and would not meaningfully change our findings. We emphasize that we are releasing the complete dataset to allow the community to validate and build upon our work.

---

> > ### Author Response · Authors · 2025-11-20
> > **Response to Reviewer NcHT [2/n]**
> >
> > ### 2. Intuitiveness of results
> >
> > > This result is intuitive but expected, once one understand the context. While the work is solid, its insights are not very novel by iclr standards.
> >
> > We respectfully disagree with the characterization of our findings as 'intuitive' or 'expected' by ICLR standards. For example, Lu et al. in their work Creativity Index [1] published at this very conference last year propose to use n-gram novelty to evaluate LLM creativity, which our results provide evidence against.
> >
> > Our work is the first to empirically and rigorously test this assumption with expert annotations. Our finding that ~91% of top-quartile n-gram novel expressions are not judged as creative is thus an *unintuitive* result given the direction of prior work at ICLR. We note that other reviewers describe the findings as useful (WN29), interesting and novel (rBr1) and surprising (Z5SQ).
> >
> > [1]  Lu, Ximing, et al. "AI as Humanity’s Salieri: Quantifying Linguistic Creativity of Language Models via Systematic Attribution of Machine Text against Web Text." The Thirteenth International Conference on Learning Representations.
> >
> > ### 3. Low F1
> >
> > > LLM as a judge's F1 performance is overly low. Some more insights would be helpful. The current statement "it suggests it may be stemming from failure to recognize non-pragmatic expressions" is unconvincing.
> >
> > We thank the reviewer for this point as it will help us better contextualize the results presented in the LLM-as-a-Judge (LLM-J) section. The task at hand is very difficult, and achieving 100% would mean perfectly modeling every annotator’s (subjective) judgement. Hence, our goal is not to reach the maximum possible F1 but rather compare the models and check if there is any useful signal in their judgments.
> >
> > To better contextualize the performance, we decided to add an empirical random baseline to both the novelty and pragmaticality tasks to show that the models indeed contain a useful signal on expert preferences. For each passage, we sampled n (number of expressions to extract) expressions of length m, where n and m are both sampled from negative binomial distributions, with parameters estimated from human annotations using the method of moments.
> >
> > - For novelty, this random F1 score is 9.6 (95% CI is [5.3, 14.0]). This means that reasoning models outperform this empirical baseline by roughly *4 times (Few-shot GPT-5 Reasoning model F1 ~ 41.3)*.
> > - For pragmaticality, the random F1 score is 2.3 (95% CI: [0.0, 4.9].
> >     - Notice it is much lower than for novelty, indicating that this is inherently a harder task.
> >     - Note that reasoning models outperform this baseline by roughly *6 times (Few-shot GPT-5 Reasoning model F1 ~ 13.5)*.
> >
> > How can we be sure that the novelty scores are useful in practice? We find that LLM-J novelty scores provide a promising signal on entirely unseen datasets:
> > - They are a statistically significant predictor of expert preference (p=0.014) (lines 376-378)
> > - They show alignment with OOD crowd preferences (p=0.054, Appendix D)
> > - They outperform n-gram based Creativity Index (Appendix E) in predicting expert preferences.
> > - For LLM-J pragmaticality scores, we do find alignment with crowd preferences (p = 0.020) but not with expert preferences (see discussion in Appendix D).
> >
> > The failure to recognize non-pragmatic expressions has been observed in the generation setting (hence the negative relationship between n-gram novelty and pragmaticality), so it was mentioned as a possible explanation for low performance on non-pragmatic expression identification in the classification setting.

---

> > > ### Author Response · Authors · 2025-11-20
> > > **Response to Reviewer NcHT [3/3]**
> > >
> > > ### 4. Fine-tuned model performance
> > > > Similarly, finetuned pragmatic or novelty identification models perform poorly, with error bars swing to as bad as x = 0. A further invesitgation on why, or finetuning larger model might be needed.
> > >
> > > We note that besides the most popular models such as Qwen, LLama and Olmo, we have fine-tuned a large model (via OpenAI API): GPT-4.1, which according to rumors is an 1.8T parameter MoE model. However, to further address the concern we decided to also fine-tune Gemini 2.5 Pro model:
> > > - For novelty, F1 is at 39.0 (95% CI: [33.8, 44.2]), significantly outperforming the baseline and comparable to reasoning models that use much more inference compute and memory.
> > > - For pragmaticality, F1 is at 8.8 (95% CI: [4.8, 12.8]), similarly outperforming the baseline and close to the few-shot reasoning model performance.
> > >
> > > It does appear that only larger models benefit from fine-tuning, with CIs not swinging to 0 for either GPT-4.1 or Gemini-2.5-Pro, and for Gemini-2.5-Pro outperforming random baseline.
> > >
> > > Due to high reasoning intensity, standard fine-tuning (training on passage -> expressions tuples) may just not be the optimal way to train for this task. Future work can explore techniques such as synthetic reasoning traces (to train reasoning models) or RL training based on our expert-annotated data.
> > >
> > > We thank the reviewer for the productive feedback which will help us improve our paper! If any concerns remain unaddressed or there are any questions, please let us know. We will be uploading a revision of the paper incorporating your feedback shortly, please let us know if there are any more comments!

---

### Official Review · Reviewer_rBr1 · 2025-11-01

**Soundness:** 2
**Presentation:** 3
**Contribution:** 3
**Rating:** 6
**Confidence:** 3

**Summary:**

This paper re-examines n-gram novelty metrics which have been previously used as proxies for creativity. Specifically, they collect a pool of annotators to measure the creativity of llm-generated text and human gold text via close reading, finding that while n-gram novelty correlates with creativity, it also corresponds to lower pragmaticality compare to human texts. Their findings suggest defining better metrics for creativity that incorporate multiple factors.

**Strengths:**

* The paper is well-written and easy to follow
* A large amount of effort went into collecting a diverse set of annotations across many annotators and passages. The definition and task for annotators are clearly defined. In general, Section 2 is strong. The experiments are designed well and a lot of thought was put into them.
* The findings about n-gram novelty relation to creativity (human judgement) are interesting and novel. There is a lot of analysis done that goes into specific details around this behavior which are interesting, like the decreasing pragmatism.
* A variety of models are tested

**Weaknesses:**

* Minor: Formatting on line 2 is odd " (Handa et al., 2025) (Anthropic) and (Chatterji et al., 2025) (OpenAI)"
* The main study comparing human and text generation is only based only on one source - The New Yorker. While this source is high-quality, it would be nice to compare LLM/Human texts in other domains to see how well n-gram novelty findings hold.
* Minor: The findings could be stronger by including more models to analyze (in addition to the task mentioned above).
* A conceptual limitation is that the paper assumes creative expressions must be both sensical and pragmatically appropriate. While this is reasonable for narrative prose, some creative language intentionally violates pragmatic norms (e.g., surreal or poetic writing). Thus, the finding that high n-gram novelty often corresponds to non-pragmatic text may not be a fundamental flaw of the metric on its own.

**Questions:**

* Given the findings, what is an alternative metric for creativity?

---

> ### Author Response · Authors · 2025-11-20
> **Response to Reviewer rBr1 [1/1]**
>
> We thank the reviewer for their assessment of our work as well-written, with well-designed experiments and interesting and novel findings. We will implement the formatting suggestions to improve the paper. We hope to address some of the concerns below:
>
> ### 1. The New Yorker as a Source
>
> > The main study comparing human and text generation is only based only on one source - The New Yorker. While this source is high-quality, it would be nice to compare LLM/Human texts in other domains to see how well n-gram novelty findings hold.
>
> We focus on the fiction domain because creativity is its main objective, as opposed to journalism or technical writing where factuality is equally (if not more) important. Creativity judgments for poetry have a higher degree of subjectivity than fiction [1,2], requiring more raters with high expertise to achieve a high-powered study. For these reasons, we turned to professionally edited, ‘gold-standard’ fiction from The New Yorker. While it is true that The New Yorker is a single source, we expect there to be substantial variation in our samples, since authors in our sample are diverse, and include several Nobel prize-winning authors.
>
> ### 2. More models to analyze
>
> > Minor: The findings could be stronger by including more models to analyze (in addition to the task mentioned above).
>
> We have also included a fine-tuned version of Gemini-2.5-Pro in our analysis (see response to reviewer NcHT), and will add Calude-4.5, Gemini-3, and GPT-5.1
>
> ### 3. Pragmaticality definition encompassing deliberate violations of pragmatic norms
>
> > A conceptual limitation is that the paper assumes creative expressions must be both sensical and pragmatically appropriate. While this is reasonable for narrative prose, some creative language intentionally violates pragmatic norms (e.g., surreal or poetic writing). Thus, the finding that high n-gram novelty often corresponds to non-pragmatic text may not be a fundamental flaw of the metric on its own.
>
> Our pragmaticality definition focuses on making sense in the context of the passage, which allows for deliberate violations of pragmatic norms. Even in surreal writing, expressions still have to make sense in the context of the passage – that would be the key distinction between a literary work and random gibberish. Therefore, we believe our operationalization of “pragmaticality” does incorporate the cases that you describe.
>
> Note also that for humans, n-gram novelty does not correlate with non-pragmaticality, unlike for open-source LLMs (Figure 3, lines 225-234). The key flaw of the n-gram novelty metric is that the vast majority of the expressions identified as highly novel were not actually judged as novel by humans (it is not sufficient to just be pragmatic). For example, even among the top-quartile n-gram novelty expression ratings that were also judged sensical and pragmatic, 89% (95% Wilson CI [0.88, 0.90]) are judged as non-novel.
>
> ### Answers to questions
>
> > Given the findings, what is an alternative metric for creativity?
>
> We agree that scalable evaluation of creativity is an important open problem. As we demonstrate in our work, the current scalable approach based on n-gram novelty is conceptually and practically flawed. Our primary contribution is to rigorously define what a correct metric should measure based on prior work in psychology of creativity.
>
> LLM-as-a-Judge (LLM-J) few-shot prompted with our data presents a proof-of-concept that this theoretically-grounded definition can be successfully operationalized. Specifically, we find that LLM-J novelty scores provide a promising signal on unseen datasets:
> - They are a statistically significant predictor of expert preference (p=0.014) (lines 376-378)
> - They show alignment with OOD crowd preferences (p=0.054, Appendix D)
> - They outperform n-gram based Creativity Index (Appendix E) in predicting expert preferences.
>
> Further work can explore techniques such as synthetic reasoning traces (to train reasoning models) or RL training based on our expert-annotated spans to improve creativity annotation performance.
>
> In addition, we show that Writing Quality Reward Models introduced in prior work are also significantly predictive of creativity judgements (297-305).
>
> We thank the reviewer for the productive feedback which will help us improve our paper! If any concerns remain unaddressed or there are any questions, please let us know. We will be uploading a revision of the paper incorporating your feedback shortly, please let us know if there are any more comments!
>
> ### References
>
> [1] Kaufman, James C., et al. "A comparison of expert and nonexpert raters using the consensual assessment technique." Creativity Research Journal 20.2 (2008): 171-178.
>
> [2] KAUFMAN, J.C., BAER, J. and COLE, J.C. (2009), Expertise, Domains, and the Consensual Assessment Technique. The Journal of Creative Behavior, 43: 223-233. https://doi.org/10.1002/j.2162-6057.2009.tb01316.x

---

### Official Review · Reviewer_WN29 · 2025-11-02

**Soundness:** 3
**Presentation:** 4
**Contribution:** 3
**Rating:** 8
**Confidence:** 4

**Summary:**

The paper tests whether n-gram novelty is predictive of creativity. The taxonamize creativity into sensibility and pragmatically and build an expert-annotated dataset to study this (~8000 samples are annotated). They conclude n-gram novelty is not a reliable predictor of creativity and worse of higher n-gram novelty is also correlated with loss of sensicality.

**Strengths:**

- The study between the n-gram novelty and creativity judgements from experts engenders a useful conclusion.  Sensibility vs pragmatically distinction is also very interesting;
- The resulting corpus is substantial, providing scope for running large scale analysis for future work.

**Weaknesses:**

- Deviation from layman perception of high quality creativity versus an expert perception of high quality creativity: Since the definitions of creativity are expert curated the claims in the paper seem very plausible; However, it is unclear if the authors expect these conclusions to hold with less expert audiences who perceive creative texts through less varying rubrics (some of which may include allegedly spurious factors like novel n-grams)

- Conditional dependence of sensicality/pragmaticality on the length of the snippet: It is unclear if the decision of the in-context sensicality or pragmaticality can vary if additional context is provided to the annotators. I understand that this is non-trivial to ablate due to the substantial variation in the induced cognitive burden to annotators but an ablation for this even a small control experiment that maintains the stability of the judgement with varying lengths over the snippet will help confirm the legitimacy,  of especially the sensible judgements.

**Questions:**

- In Ln 217, you mention that ‘Instances that were labeled as not sensical were removed to focus on cases where expressions do not make sense in the context’; Will this not alter the logical flow of the entire text ?
- You mention in Appendix D, that the pragmaticality scores consistently know a significant negative effect with the crowd preferences - Couldn’t this be indicative of the pragmaticality definition being ill-suited/divergent from what a naive-user/lay crowd prefers ?

---

> ### Author Response · Authors · 2025-11-20
> **Response to Reviewer WN29 [1/2]**
>
> We thank the reviewer for their assessment of our work as useful, interesting and substantial and for the insightful questions/comments!
>
> ### 1. Layman/expert perception
>
> > Deviation from layman perception of high quality creativity versus an expert perception of high quality creativity: Since the definitions of creativity are expert curated the claims in the paper seem very plausible; However, it is unclear if the authors expect these conclusions to hold with less expert audiences who perceive creative texts through less varying rubrics (some of which may include allegedly spurious factors like novel n-grams)
>
> In creativity psychology literature, it is recommended to focus on expert assessments in creativity [1, 2, 3], and experts are similarly preferred in LLM creativity evaluation [4]. However, to directly address your point, in Appendix D, we present a relevant discussion. While LLM-as-a-Judge novelty scores align with both expert and crowd preferences, pragmaticality scores only align with crowd preferences. This may indicate a divergence between expert and crowd preferences on nonpragmatic expressions in writing.
>
> ### 2. Dependence of sensicality/pragmaticality on context length
>
> > Conditional dependence of sensicality/pragmaticality on the length of the snippet: It is unclear if the decision of the in-context sensicality or pragmaticality can vary if additional context is provided to the annotators. I understand that this is non-trivial to ablate due to the substantial variation in the induced cognitive burden to annotators but an ablation for this even a small control experiment that maintains the stability of the judgement with varying lengths over the snippet will help confirm the legitimacy, of especially the sensible judgements.
>
> We thank the reviewer for this valid point. We will address your point by separating the concerns regarding sensicality (making sense as a standalone expression) and pragmaticality (making sense in context):
>
> - For sensicality, judgements should not vary based on context as we explicitly focused on expression making sense standalone. For example, our paper's non-sensical example "he tended at cloud finger" would not become sensical with more context.
>
> - For pragmaticality, when selecting excerpt passages for annotation, we ensured that no additional context was necessary (for example, by selecting the very beginning of the full stories). Since passages were self-contained, we do not expect there to be any variation in pragmaticality ratings if provided with more content. As we report, the free-marginal multirater kappa ($\kappa_{free}$) for pragmaticality was 0.726. This high agreement indicates that the 400-word passage contained sufficient context for professional writers to reliably perform the close reading task.
>
> ### Answers to Questions:
>
> > In Ln 217, you mention that ‘Instances that were labeled as not sensical were removed to focus on cases where expressions do not make sense in the context’; Will this not alter the logical flow of the entire text ?
>
> They were only removed for the linear modeling stage. For this analysis, the logical flow of the text does not matter, as essentially the input to the linear model is simply the expert judgement, expression’s n-gram novelty score, and annotator & passage id (to control for topic and annotator variation), see eq. in lines 188-191. For the LLM-as-a-Judge analysis, where the passage text is provided, we do not remove any expressions.
>
> > You mention in Appendix D, that the pragmaticality scores consistently know a significant negative effect with the crowd preferences - Couldn’t this be indicative of the pragmaticality definition being ill-suited/divergent from what a naive-user/lay crowd prefers ?
>
> Appendix D analyzes the LLM-as-a-Judge novelty and pragmaticality scores, which are not perfect estimates of the expert scores. Since we saw low overall performance on non-pragmatic expression identification, it is likely those scores are not reflective of the expert preferences, so it cannot suggest that the definition of pragmaticality is ill-suited. However, this could suggest that crowd preferences align more with LLM pragmaticality preferences, but the expert ones do not – this means we should not use crowd preferences to optimize the models for expert-level writing.
>
> We thank the reviewer for the productive feedback which will help us improve our paper! If any concerns remain unaddressed or there are any questions, please let us know. We will be uploading a revision of the paper incorporating your feedback shortly, please let us know if there are any more comments!

---

> > ### Author Response · Authors · 2025-11-20
> > **Response to Reviewer WN29 [2/2]**
> >
> > ### References
> > [1] Amabile, T. M. "The Social Psychology of Creativity: A Consensual Assessment Technique." Journal of Personality and Social Psychology 43, no. 5 (November 1982): 997–1013.
> >
> > [2] Kaufman, James C., et al. "A comparison of expert and nonexpert raters using the consensual assessment technique." Creativity Research Journal 20.2 (2008): 171-178.
> >
> > [3] KAUFMAN, J.C., BAER, J. and COLE, J.C. (2009), Expertise, Domains, and the Consensual Assessment Technique. The Journal of Creative Behavior, 43: 223-233. https://doi.org/10.1002/j.2162-6057.2009.tb01316.x
> >
> > [4] Chakrabarty, Tuhin, et al. "Art or artifice? large language models and the false promise of creativity." Proceedings of the 2024 CHI Conference on Human Factors in Computing Systems. 2024.

---

### Author Response · Authors · 2025-12-03
**Discussion Summary**

# Summary Comment

We thank all reviewers for their thoughtful feedback. Before focusing on summarizing how we addressed all the issues raised by the reviewers, we highlight the points of positive consensus of the 4 reviewers for our paper:
- **The usefulness / novelty of our conclusions** about n-gram novelty’s relation to creativity (WN29, rBr1, Z5SQ) and pragmatism (rBr1).
- **The quality of the resulting corpus:** substantial (WN29), requiring large effort and rigorous, high-quality expert annotation (rBr1, NcHT, Z5SQ). Reviewers particularly highlighted our annotation protocols as clear (rBr1), elaborate and well-defined (NcHT), and achieving good inter-annotator agreement (Z5SQ).
- **the presentation** as well-written and motivated (rBr1, NcHT, Z5SQ)
- **the thoroughness** of statistical analysis (rBr1, NcHT) and the variety of tested models (rBr1)

We addressed all reviewers’ concerns in the following way:

- Concerns regarding the choice of fiction as a domain and New Yorker as a source (rBr1, Z5SQ) were addressed by clarifying the diversity of the source as well as our scope in evaluating creativity (and not competing objectives such as factuality or clarity present in other genres such as journalism and technical writing).

- Concerns regarding the scale of the study (NcHT, Z5SQ) were addressed by emphasizing that the current scale allows to support our core claims with sufficient statistical significance:

    - **Inadequacy of n-gram novelty**: Among top-quartile n-gram novel expression ratings, 91% were not judged creative *(Wilson 95% CI: [0.90, 0.92]; n = 3,995 expression ratings)*. As the confidence interval is already very narrow, additional passages would not affect the substantive claim that n-gram novelty has a high false-positive rate.
    - **Negative correlation between n-gram novelty and pragmaticality**: the regression yields β = -0.48, p < 0.001, indicating a statistically and practically significant effect.
    - **Human vs. LLM creativity rates**: All models show significantly lower creativity prevalence than humans (all p < 0.03, all 95% CIs < 1; Table 2), note we corrected for multiple comparisons via Dunnett’s method.
    - **Writing quality reward model correlations**: WQRM scores predict both creativity and pragmaticality (OR 1.27-1.32, p < 0.001).
    - **LLM-as-a-judge alignment**: LLM-as-a-judge novelty scores predict expert preferences (OR = 1.88, p = 0.014).


    - In addition, we provide a comparison to a vast number of work with expert annotators far below our annotator pool size of n=26 highly qualified writers. We also highlighted the high number of resulting observations:

        - **7,542 expression ratings** across sensicality, pragmaticality, and novelty (with written justifications)
        - **226 expert-highlighted creative expressions** with written justifications
        - **26 professional writers** (larger than typical expert-annotation studies)

- We answered WN29 and Z5SQ’s questions on scalable creativity evaluation by pointing out that LLM-as-a-Judge novelty scores provide a promising signal on *unseen* datasets:
    - They are a statistically significant predictor of expert preference (p=0.014) (lines 376-378)
    - They show alignment with OOD crowd preferences (p=0.054, Appendix D)
    - They outperform n-gram based Creativity Index (Appendix E) in predicting expert preferences.


- We addressed NcHT’s concerns on LLM-Judge performance by contextualizing results with an empirical random baseline and testing and fine-tuning additional models.


- We addressed all additional/individual concerns by providing clarifications to the reviewers with appropriate citations.

While we didn’t get a response from the reviewers due to the shortened discussion period, we believe all concerns are addressed and no new issues were brought up.

---

### Meta-Review · Area_Chair_3o7B · 2026-01-12

**Summary:**

The four reviewers all recognized the interesting contribution of this work, and have provided a list of constructive critiques of the work. Some of their concerns are as follows:

•	Concerns on the differences between layman’s perception of creativity quality vs experts’ perception, and the impact of such differences on the conclusion of the paper
•	Concerns on the stability of sensicality/pragmatic evaluation w.r.t. the context length, and the lack of a controlled ablation study on this impact
•	Limited data source (only New Yorker) in conducting the analysis, which may limit the generality of the conclusion
•	Potential conceptual limitation on assuming that creative expressions must be both sensical and pragmatically appropriate
•	Limited insights
•	The need more detailed analysis, such as LLM’s low performance
•	The scale of the study is relatively small
•	The computation of n-gram novelty metrics as shown may not be accurate, potentially jeopardizing the conclusion

The authors have done a great job in answering those concerns with clear explanation, directly addressing some of the concerns with rational. Most of the concerns were addressed well.

Given the relatively high scores and enthusiasm seen from most reviewers for the work’s contribution, I’d recommend an accept of this submission.

**Reviewer Concerns:**

The four reviewers raised a range of issues about this work, such as

•	Concerns on the differences between layman’s perception of creativity quality vs experts’ perception, and the impact of such differences on the conclusion of the paper
•	Concerns on the stability of sensicality/pragmatic evaluation w.r.t. the context length, and the lack of a controlled ablation study on this impact
•	Limited data source (only New Yorker) in conducting the analysis, which may limit the generality of the conclusion
•	Potential conceptual limitation on assuming that creative expressions must be both sensical and pragmatically appropriate
•	Limited insights
•	The need more detailed analysis, such as LLM’s low performance
•	The scale of the study is relatively small
•	The computation of n-gram novelty metrics as shown may not be accurate, potentially jeopardizing the conclusion

But the authors have carefully addressed most of them. Though none of the reviewers responded to the authors’ comments, I found the authors’ answers are reasonable, including some of the work’s limitation (such as limited data source and the focus on fiction domains).

**Reviewer Scores:**

None reviewers would have changed their scores.

---

### Decision · Program_Chairs · 2026-01-26

Accept (Poster)